# Monitoring and Recording Changes in Natural Landscapes: A Case Study from Two Coastal Wetlands in SE Italy

**Valeria Tomaselli** [1,*], **Giuseppe Veronico** [2] **and Maria Adamo** [3]

1   Department of Biology, University of Bari "Aldo Moro", via Orabona 4, 70126 Bari, Italy
2   National Research Council-Institute of Biosciences and BioResources, CNR-IBBR, via Amendola 165/A, 70126 Bari, Italy; veronico81@libero.it
3   National Research Council-Institute of Atmospheric Pollution Research, CNR-IIA, via Orabona 4, 70126 Bari, Italy; adamo@iia.cnr.it
*   Correspondence: valeria.tomaselli@uniba.it; Tel.: +39-080-5442159

**Abstract:** This study analyzed and evaluated the changes that occurred in two coastal wetlands, characterized by complex and fragmented landscape patterns, in Southern Italy, which were monitored over a period of seven years from 2007 to 2014. Furthermore, the performances of two Land Cover (LC) and habitat taxonomies, compared for their suitability in mapping the identified changes, were assessed. A post-mapping method was adopted to detect the habitat/LC changes that occurred in the study period. Various changes were identified, both inter-class changes (class conversions) and intra-class changes (class modifications), and quantified by means of transition matrices. Conversions were easily mapped, while the modification mapping depended on the taxonomy adopted: the Land Cover Classification System (LCCS) allowed the detection of morpho-structural changes in woody vegetation, but the European Nature Information System (EUNIS) showed a higher thematic resolution for the salt marsh types. The detected changes were related to specific impacts, pressures and underlying factors. Landscape indices highlighted different trends in landscape richness and complexity in the two sites. Changes are occurring very quickly in the observed coastal sites and the ongoing dynamics are strictly related to their small size and complexity. For effective monitoring and detection of change in these environments, the coupling of EUNIS and LCCS is suggested.

**Keywords:** monitoring; landscape changes; habitat; land cover; vegetation; coastal wetlands

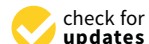



## 1. Introduction

Effective management policies for coastal wetlands require consistent monitoring procedures, as well as awareness of the ongoing socioeconomic dynamics in the geographical area under consideration. Having a knowledge framework, as comprehensive as possible, with regard to the landscape composition and the underlying ecological processes, as well as pressures and threats, is an essential starting point for the planning of effective management measures. In such a framework, Land Cover (LC) and habitat monitoring at the landscape level is a basic prerequisite [1–3]. The assessment of changes in landscape ecological elements has been identified as an important topic for future research (2007 IALE World Congress) and requires stringent procedures to ensure that the recorded differences represent real changes and not distortions due to discordances between observers or recording techniques [4,5].

A crucial step is the representation of the detected changes, which may occur in terms of both conversion and modification. A conversion, in general, is easy to map by simply shifting from one class to another, while a modification—that is, a transformation within a class—is a challenging issue to address in mapping procedures. In this sense, the outcome of the mapping process and the chance of detecting and representing changes are closely related to the LC or habitat taxonomy used; that is, to its structure, semantics and especially its level of thematic resolution [6–9].

Coastal wetlands are areas of high biological diversity that play a key role in maintaining and enhancing a wide range of ecosystem services, such as improving water quality, equilibration of the water cycle, carbon sequestration, providing a natural habitat for migratory birds and recreation [10,11]. Land claim, agricultural intensification and hydrological modifications are the main drivers of change for these habitats, but urbanization and the introduction of alien species also represent important threat factors [12]. In the Mediterranean region, about a half of the wetlands have been lost in the course of the 20th century [13,14]. Moreover, the Mediterranean territories, included coastal areas and wetlands, are characterized by a wide variety of historical land uses, which are expressed in a complex, diverse and highly fragmented landscape pattern [15–18].

Quantifying wetland surface area and trends at a regional Mediterranean level has received increasing attention in recent decades. Many of these evaluations rely on land cover maps derived from photointerpretation or from satellite images and classified on the basis of the Corine Land Cover (CLC) taxonomy or coarser LC taxonomies [13,19,20]. Even if CLC is a valid tool for the evaluation of broad changes at large scales, it does not allow discrimination among different wetland habitat types at a detailed scale [9,13,21–23].

The LC and habitat taxonomies most commonly used in mapping procedures are limited in their ability to read all aspects of the landscape and often do not contain the full diversity of possible natural and anthropogenic types. The potential for detecting changes relies on the possibility of describing LC and habitat types also according to their structure (e.g., stratification, cover, etc.). Among the LC taxonomies, the Land Cover Classification System (LCCS) was launched by the Food and Agricultural Organization (FAO) for standardization and harmonization of land cover/land use (LC/LU) information [24,25] and was demonstrated to be the most appropriate and user-friendly framework for the harmonization of different LC taxonomies [26]. In addition, it has been successfully adopted for the translation of LC classes to habitat types in habitat monitoring procedures based on Earth Observation (EO) data [21–23]. Moreover, for its particular structure, LCCS seems to be well suited to detecting LC changes (that is, class modifications).

In this study we analyzed and evaluated the changes that occurred in two coastal sites in the region of Apulia (Southern Italy), which are characterized by highly complex and fragmented landscape patterns and high levels of anthropogenic pressures, in a time frame of seven years, from 2007 to 2014. Furthermore, we evaluated the performance of two (habitat and land cover) taxonomies, compared for their "ability" in identifying and mapping the changes that occurred in this time frame.

## 2. Materials and Methods

### 2.1. Study Areas

Two study sites belonging to the EU Natura 2000 network and located in Apulia (Southern Italy) were selected: Le Cesine (acronym used in this study: CE; SCI IT9150032, SPA IT9140003) and Saline di Punta della Contessa (acronym used in this study: SC; SCI IT9140003, SPA IT9150014). The sites cover about 810 and 210 ha, respectively (Figure 1).

These two areas were selected because they fall within the same geographical and administrative region and have many similar features: despite their limited extent, both the areas are characterized by a high diversity in habitats and vegetation types, all forming intricate ecological patterns [27,28]; in Appendix A, the complete list of habitat and land cover types characterizing the two study sites is reported. The two sites have different managing bodies and management plans. As for pressures and threats, some are common to both sites (e.g., habitat loss and fragmentation due to coastal erosion; encroachment in Mediterranean garrigues, mainly due to frequent fires; see Appendix B), while others are not (e.g., water salinization causing loss of reed bed communities in the CE site; the opposite trend, that is, reed bed rapid expansion, in the SC site is due to water eutrophication; see also Appendix B) and this also depends on the policies applied in the management of the two sites.

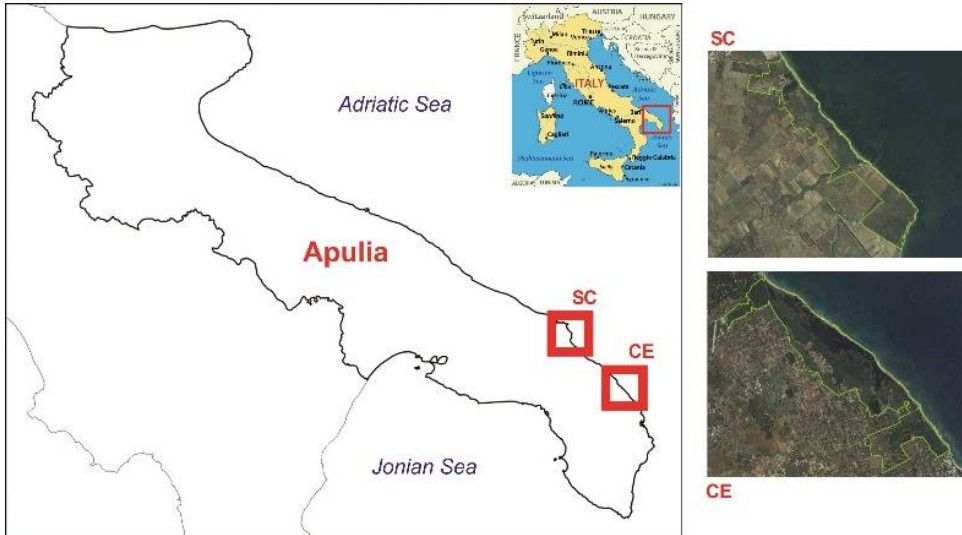

**Figure 1.** Geographical location and administrative boundaries of the two study areas.

*2.2. Mapping*

The mapping was performed by means of photointerpretation and on-site surveys undertaken in the years 2007 and 2014. This time frame corresponds to the final (fourth) habitat report (Art.17 of Habitat Directive). The thematic maps of the study site were digitized in ArcGis 10.2 from color orthophotos (2006 and 2013) in combination with topographic maps (source: SIT-Puglia, http://www.sit.puglia.it/). Natural and seminatural landscape elements were first defined as vegetation types (phytosociological units, according to the Zurich–Montpellier method; [29]) on a 1:5000 scale, which allowed the studied landscapes to be represented with 5 m resolution. The photointerpretation was carried out in three steps: first, the areas whose attribution to a specific type was certain were bordered (i.e., 100 percent certainty areas); then, the zones initially left blank (i.e., those whose classification initially seemed doubtful) were attributed to a specific type based on expert knowledge; finally, these "doubtful zones" were verified by in-field campaigns carried out in 2007–2008 and in 2014–2015. Thus, the final map had an accuracy of almost 100% [30]. Information on vegetation composition and structure, as well as agricultural practices or land use, was collected, geocoded by GPS and integrated into a geographic information system (GIS) geodatabase for accurate and detailed definition of some types.

Vegetation units were reclassified in habitat types and then in LC classes. For habitat mapping, we applied the European Nature Information System (EUNIS) [31] classification scheme (levels III and IV) as it is recognized as an important standardizing tool for habitat classification in the EU [32]. For LC classes, we referred to the Food and Agriculture Organization Land Cover Classification System taxonomy [24]. The structure and the potentialities of the LCCS for mapping in Mediterranean coastal wetlands have been explored in previous papers [9,21–23]. One of the fundamental aspects of the LCCS is that LC classes are not simply based on nomenclature (as in most LC or habitat taxonomies), but are defined by a set of independent diagnostic criteria strictly based on vegetation physiognomy and structure. Thus, each LC class is defined by a dynamic combination of classifiers and is described by two elements: (a) a Boolean formula, consisting of a string of classifiers used for class definition (e.g., A2.A5.A13.B4.C2.E5-B13.E7); and (b) the name of the land cover class (e.g., "open annual short herbaceous vegetation on temporarily flooded land"). The increase of detail in the description of a land cover feature is linked to the increase of classifiers used. As an example, a pine forest can be described as Trees (A12-A3), as Needleleaved evergreen trees (A12-A3.D2.E1) or with more complex descriptions, including details on cover, height and stratification, according to the user's requirements. Such detailed information on vegetation structure also makes possible the detection of changes in terms of modification within a specific habitat type. In fact, changes

become immediately recognizable by adding one or more classifiers or through the use of additional classifiers, even when maintaining the same class type [6].

The adopted landscape classification procedure refers to a hierarchical model with three different information levels: the vegetation unit, the habitat type and the LC type.

*2.3. Changes*

Magnitude Changes (MCs) in class area occurred between 2007 and 2014 and the corresponding Trend Percentage Changes (TPCs) were computed for each habitat class by using the following formulas [33,34]:

$$MC_i = CA_i(T2) - CA_i(T1) \tag{1}$$

$$TPC_i = \frac{MC_i}{CA_i(T1)} \cdot 100 \tag{2}$$

where *i* is the habitat considered and, for the case under study, *T*1 and *T*2 correspond to 2007 and 2014, respectively. *CA* represents the Class Area recorded for each class.

2.3.1. Transition Matrix

In this study we adopted a post-mapping method to detect the habitat/LC changes that occurred between two different dates of the study period in two independent maps. The approach provided comprehensive and detailed "from–to" habitat/LC change information. The output was a crosstabulation matrix (transition matrix) which consisted of rows (displaying habitat/LC class category for time 1) and columns (displaying habitat/LC class category for time 2). The following equation was used to calculate the matrix [35]:

$$p = \begin{bmatrix} p_{11} & \cdots & p_{1j} \\ \vdots & \ddots & \vdots \\ p_{i1} & \cdots & p_{ij} \end{bmatrix} \tag{3}$$

where $p_{ij}$ indicates the area percentage in transition from class *i* to *j*.

2.3.2. Landscape Metrics

On the basis of the thematic maps produced, landscape composition of the two sites was assessed through several spatial metrics computed using the LecoS–Land cover statistics [36] plugin of the QGIS GIS software suite, directly downloaded from the QGIS plugin hub (http://plugins.qgis.org/plugins/LecoS/).

At the class level, the following metrics were selected (Table 1): NP (number of patches), MaxPS (max patch size), MinPS (min patch size), MPS (mean patch size), PSSD (patch size standard deviation), PLAND (percentage of landscape), PD (patch density), ED (edge density), LPI (largest patch index) and SHAPE (shape index). These are considered effective in evaluating landscape composition [37–39] and have already been used in a previous survey on habitat fragmentation at these study sites [18]; in addition, we added COHESION (patch cohesion index) as an effective measure of dispersion and interspersion [38,40].

At the landscape level, the following metrics were selected (Table 2): NP (number of patches), MaxPS (max patch size), MinPS (min patch size), MPS (mean patch size), PSSD (patch size standard deviation), PRD (patch richness density), Shannon's diversity index (SHDI) and Simpson's diversity index (SIDI).

**Table 1.** List of the metrics selected for landscape analysis at the class level and their description.

| Index | Symbol | Description |
|---|---|---|
| Number of patches | NP | # number of patches at class level |
| Mean patch size | MPS | Average patch area of patches at the class level (ha) |
| Patch size standard deviation | PSSD | Standard deviation in patch area of patches at the class level (ha) |
| Max patch size | MaxPS | Maximum area of patches at class level (ha) |
| Min patch size | MinPS | Minimum area of patches at class level (ha) |
| Percentage of landscape | PLAND | The percentage of the landscape comprised of a particular patch type (%) |
| Edge density | ED | Density (m/ha) of edges of a particular patch type |
| Largest patch index | LPI | Percentage of the landscape comprised of the single largest patch |
| Mean patch shape ratio | SHAPE | Normalized ratio of patch perimeter to area in which the complexity of patch shape is compared to a standard shape (square) of the same size |
| Patch density | PD | Density (#/ha) of patches at the class level |
| Patch cohesion index | COHESION | Area-weighted mean perimeter–area ratio |

**Table 2.** List of the metrics selected for landscape analysis at the landscape level and their description.

| Index | Symbol | Description |
|---|---|---|
| Number of patches | NP | Number of patches at landscape level |
| Patch richness density | PRD | Number of patch types present normalized to total landscape area (#/ha) |
| Mean patch size | MPS | Average patch area of patches at landscape level (ha) |
| Patch size standard deviation | PSSD | Standard deviation in patch area of patches at landscape level (ha) |
| Max patch size | MaxPS | Maximum area of patches at landscape level (ha) |
| Min patch size | MinPS | Minimum area of patches at landscape level (ha) |
| Simpson's diversity index | SIDI | Probability that any two cells selected at random are different patch types |

Although the use of PRD does not fully correct for the patch type–area relationship in which the number of patch types increases nonlinearly with landscape area, it facilitates comparison among different landscapes. Shannon's diversity index is a universally accepted index for diversity based on information theory. It accounts for entropy in a system: it equals 0 when the reporting units contain only one class (no diversity) and increases as the number of different classes increases and/or the proportional distribution of area among habitat types becomes more equitable. Shannon's index is strongly influenced by class richness and by rare classes. Simpson's diversity index is another popular diversity measure, which is not based on information theory. Simpson's index is calculated as 1 minus the sum of the proportional areas of each patch type squared; specifically, the value of Simpson's index represents the probability that any two cells selected at random will be different patch types. Thus, the higher the value the greater the likelihood that any two randomly drawn cells will be different patch types. Simpson's index is less sensitive to the presence of rare types and responds most strongly to changes in the proportional abundance of the most common classes [41–43].

## 2.4. Pressures and Driving Factors

In order to support the interpretation of the detected changes and to consistently describe driving factors, stresses and pressures in different sites, we referred to the classification system proposed by Salafsky et al. [44,45] and then modified by Nagendra et al. [46].

## 3. Results

### 3.1. Habitat and LC Maps

Figures 2 and 3 show the output EUNIS maps obtained in 2007 and 2014 for the SC and CE sites, respectively.

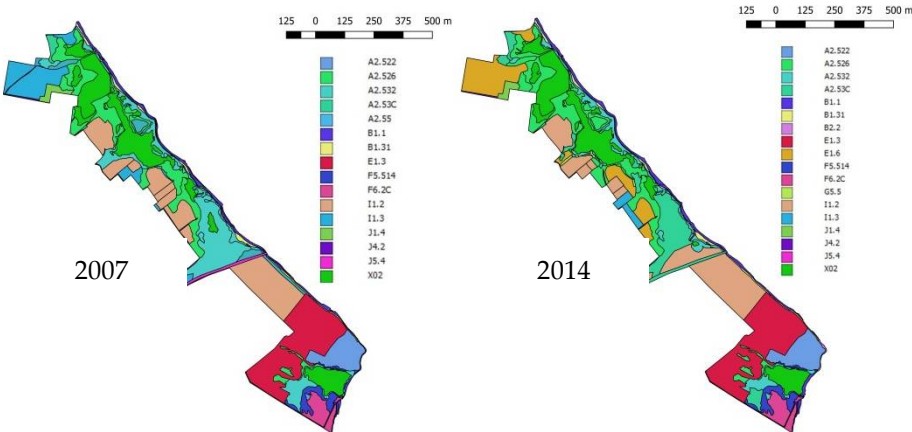

**Figure 2.** European Nature Information System (EUNIS) map of the Saline di Punta della Contessa (SC) site in 2007 and in 2014.

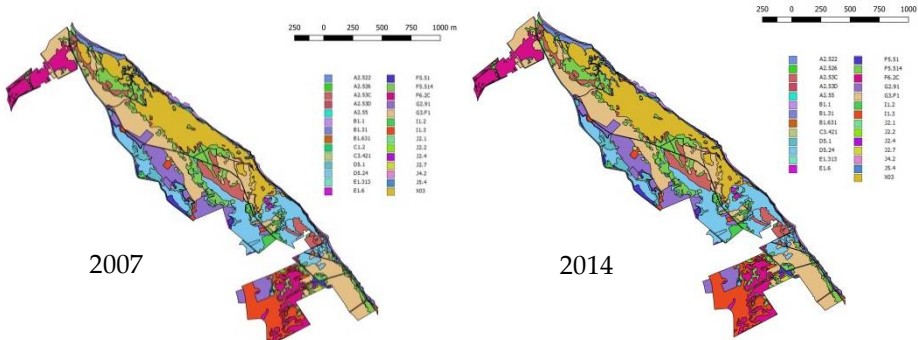

**Figure 3.** EUNIS map of the Le Cesine (CE) site in 2007 and in 2014.

The complete list of habitat types, along with LC classes, is reported in a reference table in Appendix A.

In both sites it is possible to identify a high degree of landscape heterogeneity, corresponding to both an effective natural diversity of the biotope and to a variety of land uses. Further considerations on landscape composition and configuration and on the degree of fragmentation in the two sites are provided in Section 3.2 below.

### 3.2. Changes

3.2.1. Inter-Class Changes (Class Conversion)

Habitat maps from 2007 and 2014 were analyzed to provide the habitat transition matrices related to both sites. The analysis of transition matrices revealed an overall percentage of areal changes equal to 27.5% (about 58 ha) for SC (Figure 4) and equal to 12.7% (about 102 ha) for CE (Figure 5).

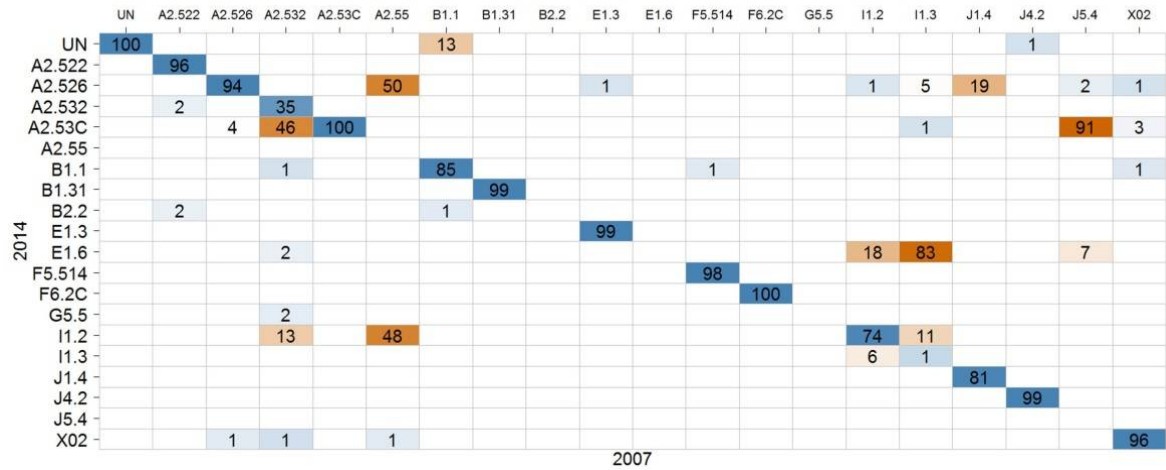

**Figure 4.** EUNIS habitat class transition matrix from 2007 to 2014 for the SC site.

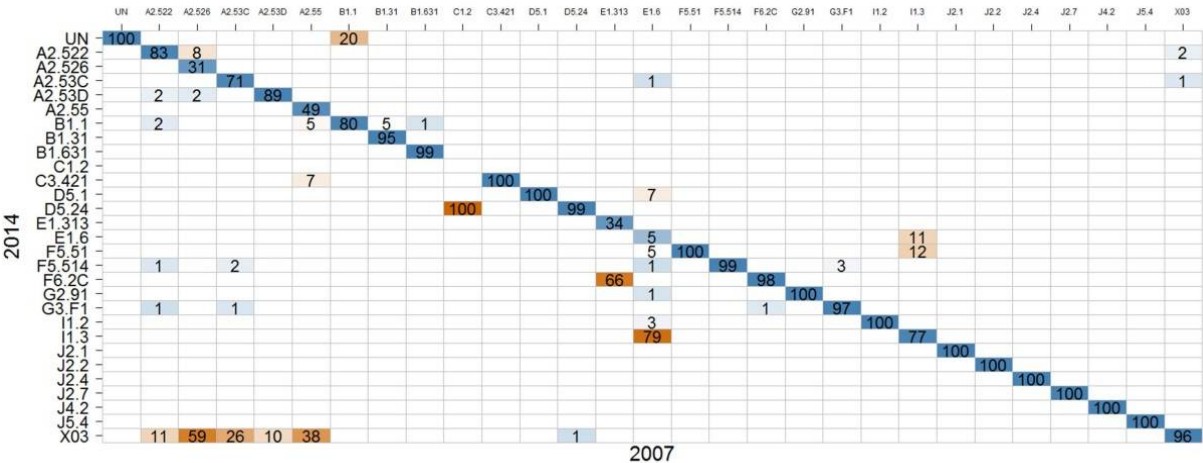

**Figure 5.** EUNIS habitat class transition matrix from 2007 to 2014 for the CE site.

As regards the SC site, one of the main changes identified concerns halo-psammophile meadows (EUNIS A2.532; trend = −63%) which were converted partly into saline reedbeds (A2.53C) and partly into cultivated areas. Another severe conversion regards pioneer salt marshes (A2.55; trend = −100%), which were entirely replaced by cultivated areas. As for the synanthropic classes, arable lands showed a drastic reduction (I1.3; trend= −85%), converted in E1.6 (subnitrophilous annual grasslands).

As regards CE, saline reedbeds (A2.53C) underwent a reduction (trend = −25%), converted into coastal lagoons (X03) (Figure 6). Sand beach driftlines and embryonic shifting dunes (classes B1.1 and B1.31, respectively) suffered a reduction due to coastal erosion. The woody vegetation, that is, Mediterranean maquis, garrigues and pine woods (classes F5.514, F6.2C and G3.F1, respectively), did not suffer relevant changes in surface but only minor fluctuations (important structural modifications cannot be represented in EUNIS). Subnitrophilous annual grasslands showed a drastic reduction (E1.6; trend = −94.5%), partly converted into D5.1 but mainly converted into I1.3 (arable lands).

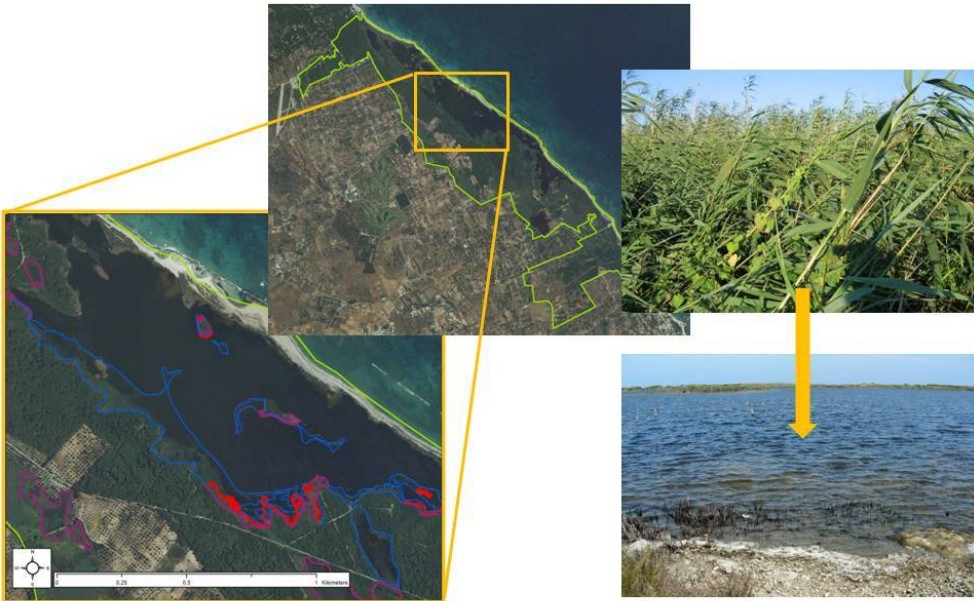

**Figure 6.** A severe reduction of reedbeds was observed in the CE site. This process was localized mostly in the central part of the protected area; in blue, the reedbeds distribution in 2007 and, in red, in 2014.

The CA recorded for each class in 2007 and in 2014, along with the MC and TPC measured from 2007–2014, are reported in Tables 3 and 4 for SC and CE, respectively.

**Table 3.** Class Area (CA) in 2007 and 2014: Magnitude Change (MC) and Trend Percentage Change (TPC) in the SC site.

| Habitat | CA in 2007 (ha) | CA in 2014 (ha) | MC (2007–2014) (ha) | TPC (2007–2014) (%) |
|---------|-----------------|-----------------|---------------------|---------------------|
| A2.522 | 9.31 | 8.96 | −0.35 | −3.79 |
| A2.526 | 25.74 | 28.27 | 2.53 | 9.82 |
| A2.532 | 32.66 | 11.97 | −20.68 | −63.33 |
| A2.53C | 1.26 | 20.39 | 19.13 | 1517.75 |
| A2.55 | 4.36 | | −4.36 | −100.00 |
| B1.1 | 5.72 | 6.28 | 0.56 | 9.74 |
| B1.31 | 3.36 | 3.35 | −0.01 | −0.36 |
| B2.2 | | 0.31 | 0.31 | |
| E1.3 | 30.48 | 30.24 | −0.24 | −0.80 |
| E1.6 | | 20.27 | 20.27 | |
| F5.514 | 4.03 | 3.96 | −0.07 | −1.65 |
| F6.2C | 4.95 | 4.95 | 0.00 | 0.00 |
| G5.5 | | 0.59 | 0.59 | |
| I1.2 | 35.09 | 33.93 | −1.16 | −3.32 |
| I1.3 | 15.79 | 2.41 | −13.38 | −84.71 |
| J1.4 | 1.73 | 1.40 | −0.33 | −18.95 |
| J4.2 | 1.14 | 1.14 | −0.01 | −0.70 |
| J5.4 | 2.01 | 0.20 | −1.81 | −90.03 |
| X02 | 32.11 | 31.28 | −0.83 | −2.58 |

**Table 4.** CA in 2007 and 2014: MC and TPC in the CE site.

| Habitat | CA in 2007 (ha) | CA in 2014 (ha) | MC (2007–2014) (ha) | TPC (2007–2014) (%) |
|---|---|---|---|---|
| A2.522 | 18.70 | 17.99 | −0.71 | −3.79 |
| A2.526 | 0.16 | 0.05 | −0.11 | −67.26 |
| A2.53C | 58.20 | 43.49 | −14.71 | −25.28 |
| A2.53D | 0.21 | 0.66 | 0.44 | 210.94 |
| A2.55 | 1.45 | 1.02 | −0.43 | −29.39 |
| B1.1 | 12.77 | 12.27 | −0.51 | −3.98 |
| B1.31 | 10.67 | 10.14 | −0.53 | −5.01 |
| B1.631 | 2.70 | 2.68 | −0.02 | −0.84 |
| C1.2 | 0.02 | | −0.02 | −100.00 |
| C3.421 | 1.29 | 1.46 | 0.16 | 12.61 |
| D5.1 | 12.59 | 16.98 | 4.39 | 34.83 |
| D5.24 | 106.67 | 105.91 | −0.75 | −0.71 |
| E1.313 | 0.90 | 0.41 | −0.49 | −54.30 |
| E1.6 | 66.17 | 3.60 | −62.57 | −94.56 |
| F5.51 | 6.77 | 10.09 | 3.31 | 48.87 |
| F5.514 | 80.85 | 85.37 | 4.52 | 5.59 |
| F6.2C | 63.66 | 62.97 | −0.69 | −1.08 |
| G2.91 | 58.81 | 59.35 | 0.54 | 0.92 |
| G3.F1 | 162.44 | 159.64 | −2.80 | −1.72 |
| I1.2 | 8.45 | 10.20 | 1.75 | 20.73 |
| I1.3 | 2.84 | 54.36 | 51.51 | 1811.02 |
| J2.1 | 0.16 | 0.16 | 0.00 | 0.00 |
| J2.2 | 0.22 | 0.24 | 0.02 | 10.04 |
| J2.4 | 0.77 | 0.77 | 0.00 | 0.00 |
| J2.7 | 1.37 | 1.46 | 0.09 | 6.93 |
| J4.2 | 9.42 | 9.55 | 0.13 | 1.36 |
| J5.4 | 6.08 | 6.08 | 0.00 | 0.00 |
| X03 | 112.65 | 126.92 | 14.26 | 12.66 |

A consideration should be noted with regard to the observed trend (TPC) values. Some salt marshes types, especially those lying at the interface between ponds and lagoons, are often prone to intra- and inter-annual fluctuations. When such types cover small surface areas, as often occurs in highly fragmented landscapes, even minor fluctuations may result in high trend (TPC) values, leading to overestimation of the process under observation.

3.2.2. Intra-Class Changes (Class Modification)

Intra-class changes (class modifications) can be detected by comparing EUNIS with LCCS maps.

The analysis of land cover transition matrices revealed an overall percentage of areal changes equal to 23.1% (about 58 ha) for SC (Figure 6) and equal to 28.9% (about 102 ha) for CE (Figures 7 and 8).

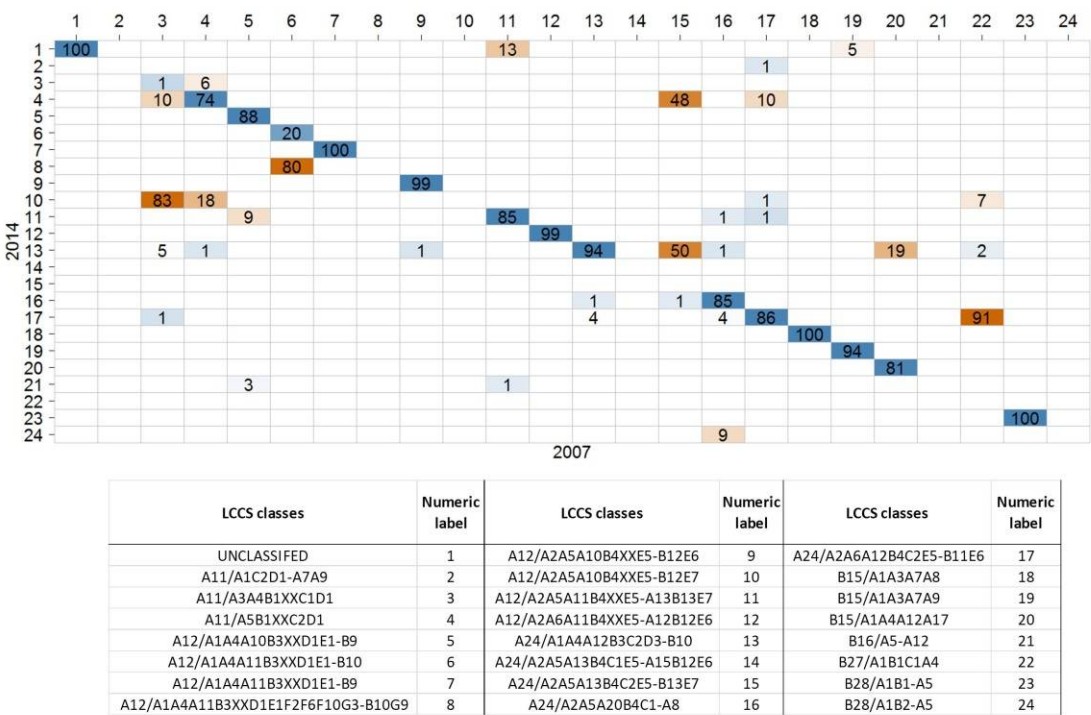

**Figure 7.** Land Cover Classification System (LCCS) habitat class transition matrix for 2007 to 2014 for the SC site.

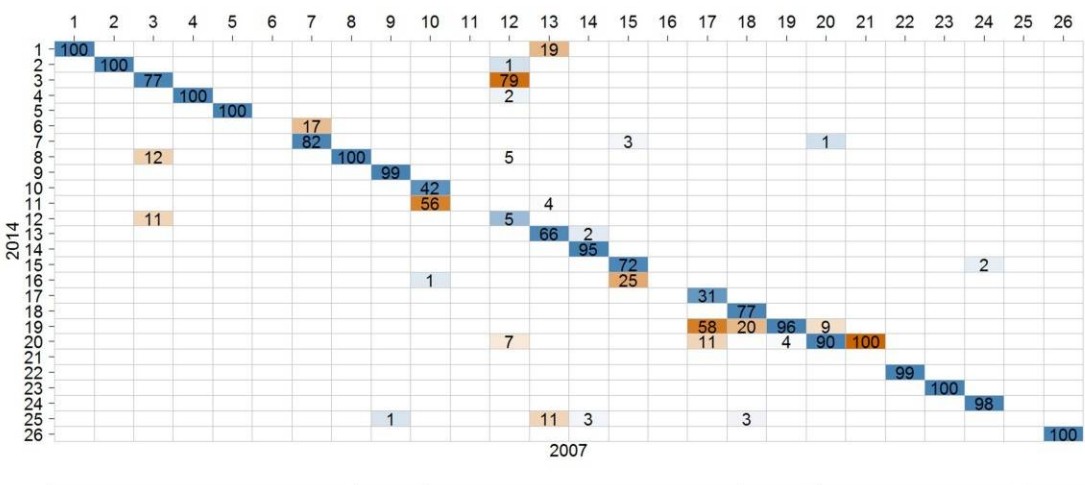

**Figure 8.** LCCS habitat class transition matrix for 2007 to 2014 for the CE site.

The LCCS identified more changes compared to the EUNIS taxonomy, especially in the CE site. This is due to the fact that, in this site, multiple intra-class changes, or habitat modifications, have occurred, corresponding to ecologically important processes that were not detected with EUNIS but that can be detected by LCCS. These changes included shrub encroachment in the garrigues (A12/A1A4A11B3XXD1E1-B10 to A12/A1A4A11B3XXD1E1F2F6F10G3-B10G9; EUNIS F6.2C), a process ongoing in both the sites. In CE, the following dynamics were also observed: pine encroachment in the Mediter-

ranean maquis (A12/A1A4A10B3XXD1E1-B9 to A12/A1A3A11B2XXD2E1F2F6F7G3-B7F8G9; EUNIS F5.514) (Figure 9); canopy forest thinning in pine forests (A12/A3A11B2XXD2E1-B6 to A12/A3A10B2XXD2E1-B6; EUNIS G3.F1) (Figure 10); alteration of the coastal drift line in terms of the ratio of vegetated to nonvegetated areas (A12/A2A5A11B4XXE5-A13B13E7 to B16/A6; EUNIS B1.1); and variations in terms the ratio of vegetated to nonvegetated areas in the natural lagoon classes (A24/A2A5A20B4C1-A8 to B28/A1B2-A5; EUNIS X02).

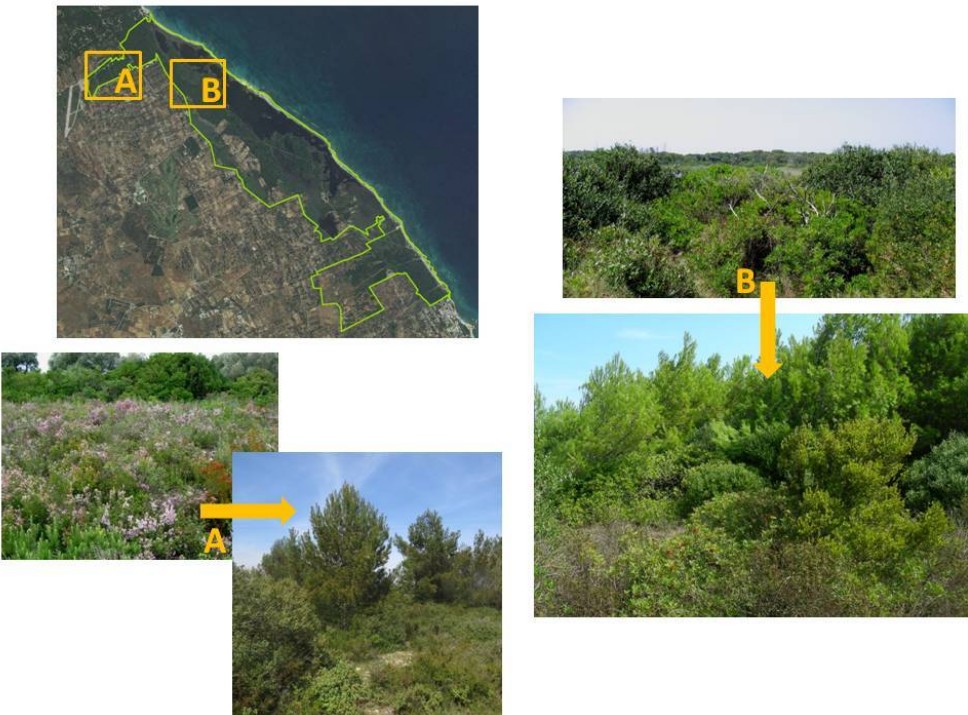

**Figure 9.** Localization of areas affected by pine encroachment in the Mediterranean garrigue (**A**) and maquis (**B**) in the CE site and photographic images depicting the ongoing process.

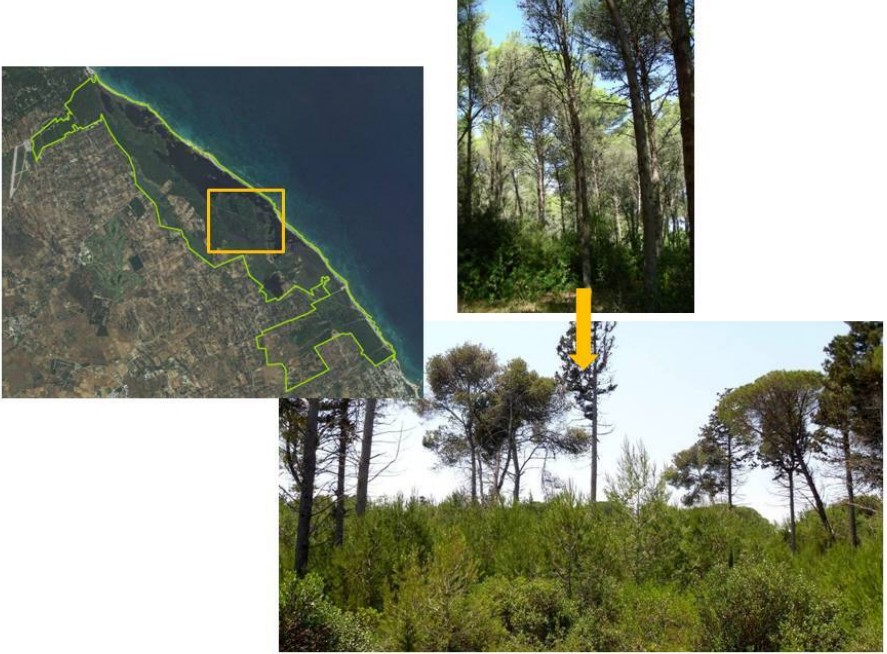

**Figure 10.** Localization of the area affected by canopy pine forest thinning in the CE site and photographic images depicting the ongoing process.

LCCS does not always have a better thematic resolution than EUNIS. The numerous different classes of helophytic vegetation, clearly distinct in EUNIS (e.g., EUNIS A2.522, A2.532, A2.53C), fall within a unique LCCS class; that is, A24/A2A6A12B4C2E5-B11E6. In the LS site these classes showed important inter-class changes (conversion; e.g., A2.532 to A2.53C) and these changes could not be detected and represented by LCCS; ultimately, LCCS does not make it possible to identify the changes between the various types of helophytic vegetation. In the CE site, the whole class A24/A2A6A12B4C2E5-B11E6 underwent a reduction, but mainly related to the reedbeds (EUNIS A2.53C to X03) and without relevant inter-class changes.

In Figure 11, the MCs of habitat and LCCS classes shared between the two sites are compared. The A2.53C (marine reedbeds) and I1.3 (arable lands) classes are the EUNIS classes that showed major MCs. In LCCS however, Mediterranean garrigues (A12/A1A4A11D3B1E1-B10) and subnitrophilous annual grasslands (or fallow lands A12/A2A5A10B4E5-B12E7) showed the main MCs. This apparently contrasting arrangement of the MC values can be related to the thematic resolution of the two taxonomies. EUNIS has a better thematic resolution for salt marshes and, in general, "wet" types; LCCS better discriminates among different morphostructural characters of woody vegetation.

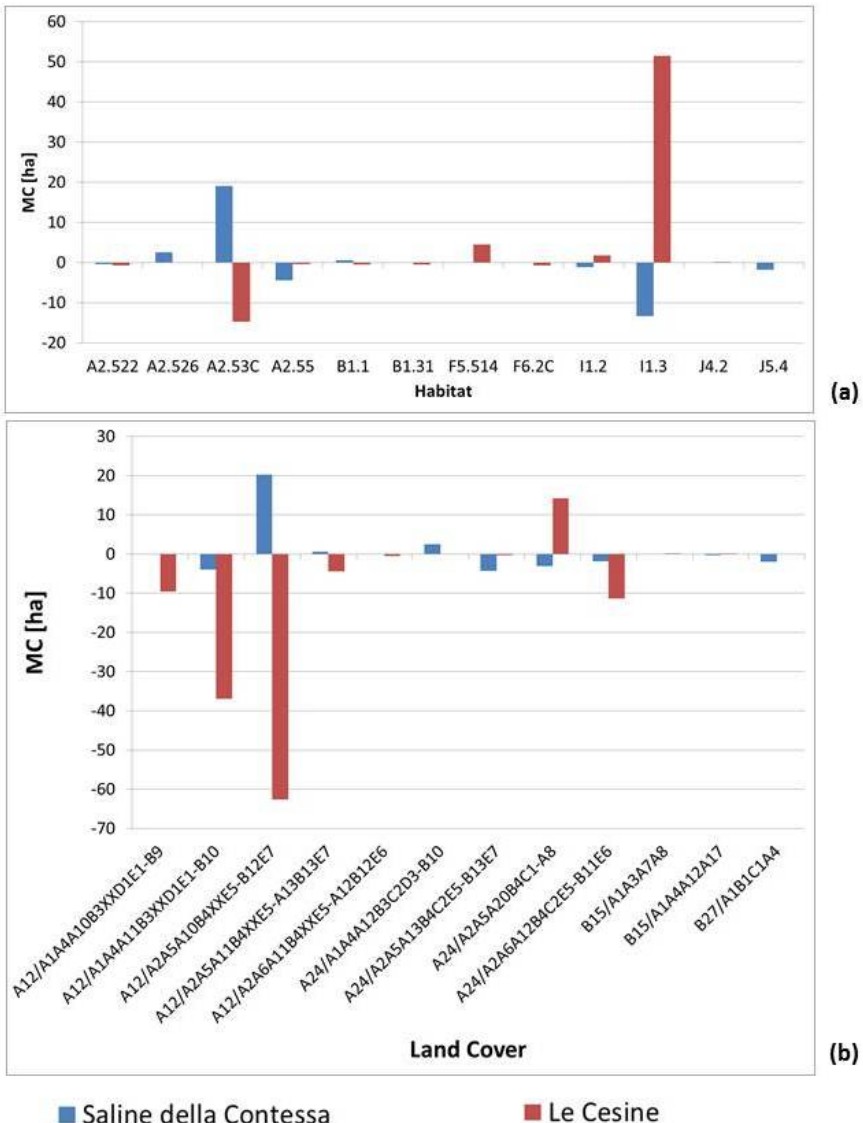

**Figure 11.** Barplots comparing the MCs for habitat (**a**) and LCCS (**b**) classes shared between the two sites.

### 3.2.3. Changes Through Landscape Metrics

Figures 12–15 graphically display the values of the landscape metrics for SC and CE, respectively, compared for the two years of observation. Different trends in different landscape metrics (graphically represented as different spikes) clearly appear and are mostly due to the high landscape heterogeneity in the two study sites; in particular, due to the different landscape configuration of the different habitat classes. As an example, in the CE site (Figures 14 and 15), class X03 (coastal lagoons) showed very high spikes in PLAND, MPS and LPI (because this class was distributed in a few large patches covering a high percentage of the landscape) but had low values in NP, SHAPE and PD (just because of its configuration); on the other hand, class F5.514 (Mediterranean maquis) showed low values for LPI, MPS and PSSD but very high values in NP, PD and ED because of its particular landscape configuration (the class consisted of numerous small irregular patches, more or less of the same size).

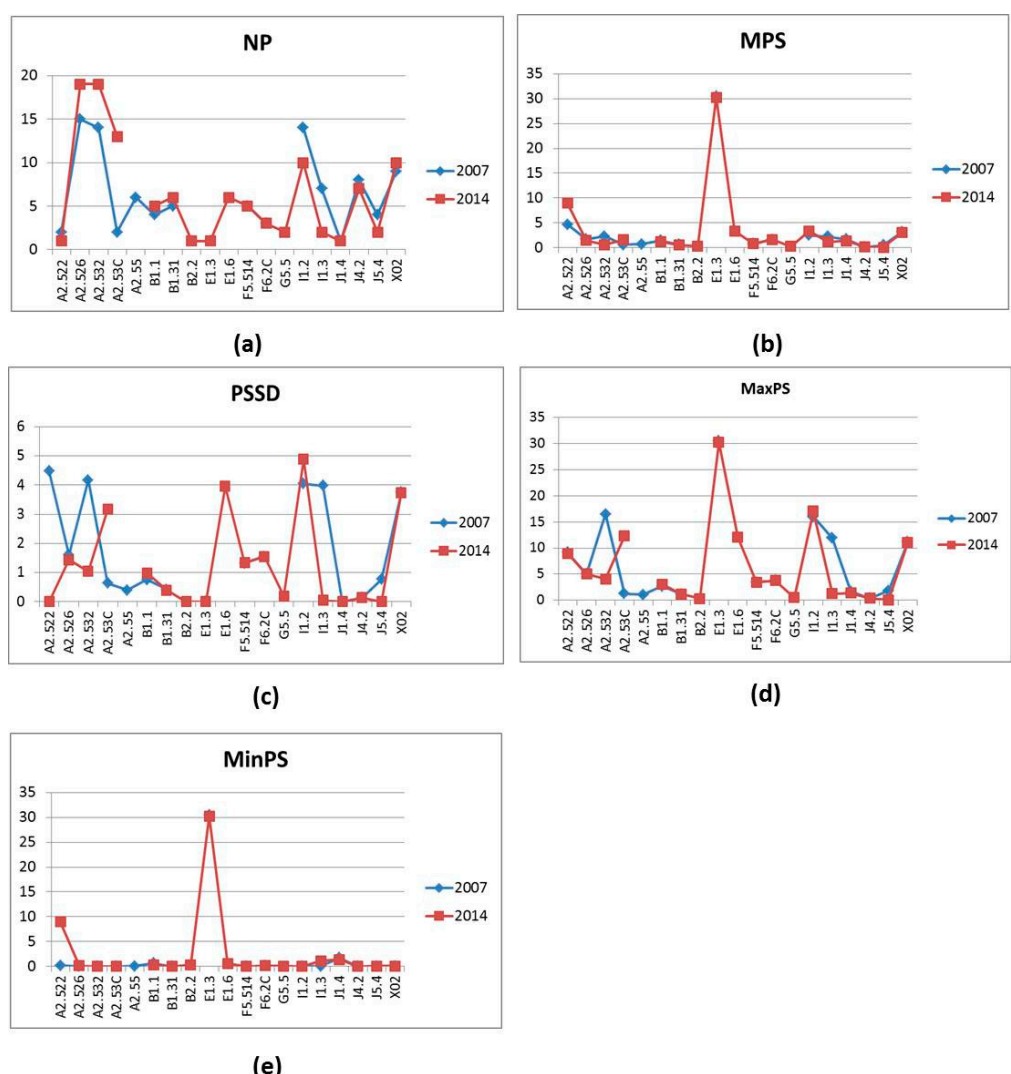

**Figure 12.** Diagrams comparing: (**a**) NP, number of patches; (**b**) MPS, mean patch size; (**c**) PSSD, patch size standard deviation; (**d**) MaxPS, max patch size; and (**e**) MinPS, min patch size at the SC site in 2007 and 2014.

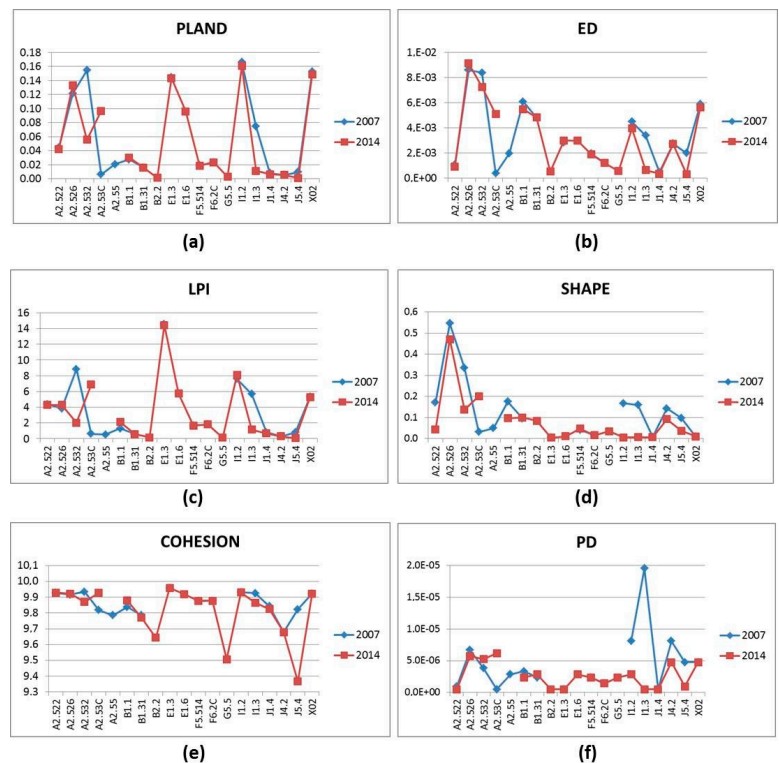

**Figure 13.** Diagrams comparing: (**a**) PLAND, percentage of landscape; (**b**) ED, edge density; (**c**) LPI, largest patch index; (**d**) SHAPE, mean patch shape ratio; (**e**) COHESION, patch cohesion index; and (**f**) PD, patch density, at the SC site in 2007 and 2014.

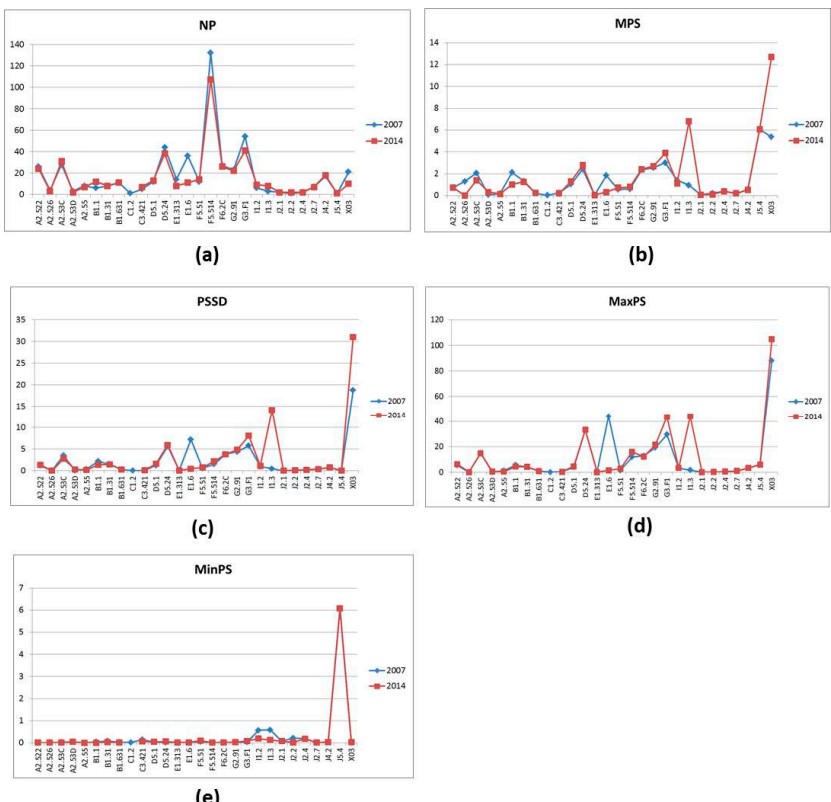

**Figure 14.** Diagrams comparing: (**a**) NP, number of patches; (**b**) MPS, mean patch size; (**c**) PSSD, patch size standard deviation; (**d**) MaxPS, max patch size; and (**e**) MinPS, min patch size at the CE site in 2007 and 2014.

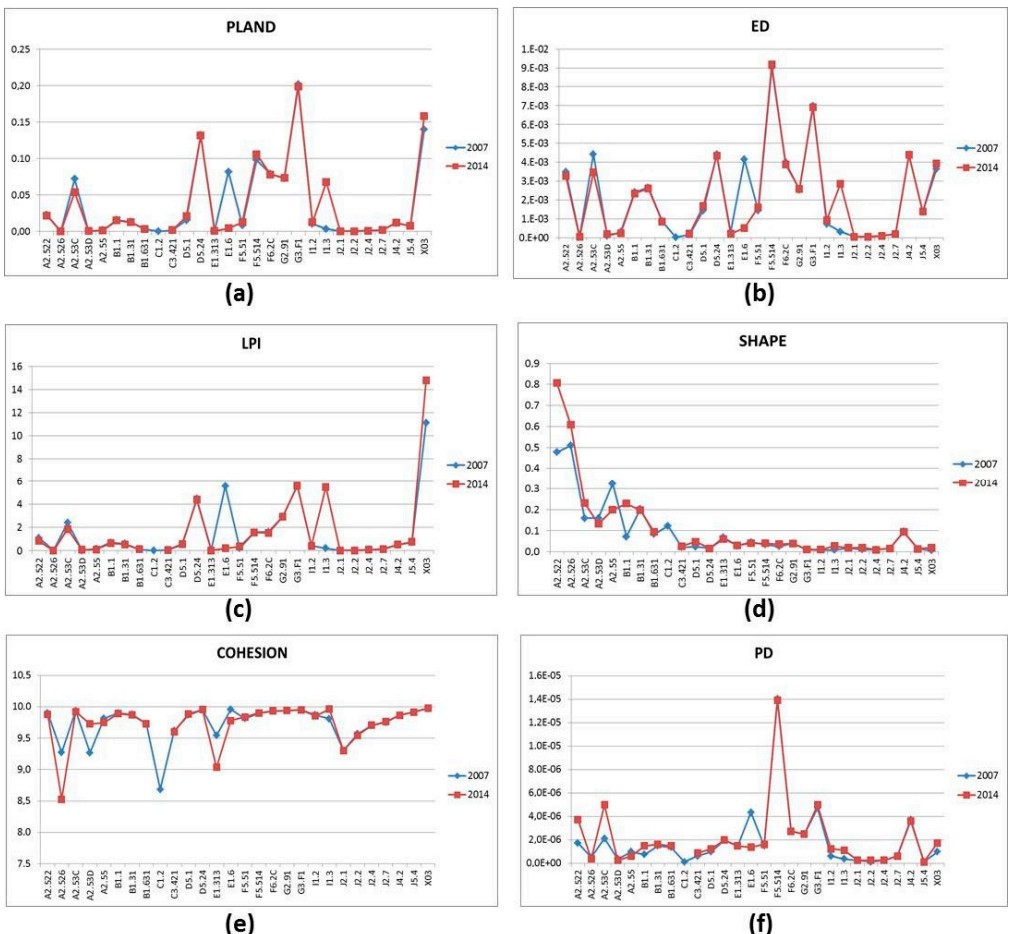

**Figure 15.** Diagrams comparing: (**a**) PLAND, percentage of landscape; (**b**) ED, edge density; (**c**) LPI, largest patch index; (**d**) SHAPE, mean patch shape ratio; (**e**) COHESION, patch cohesion index; and (**f**) PD, patch density at the CE site in 2007 and 2014.

As regards the changes detected in the SC site in the two years of observation (Figures 12 and 13), the gap—that is, the absence of values—for the A2.55 class, for which there were no results in 2014, immediately stands out. There are also evident variations, with almost all metrics, in the A2.532 and A2.53C classes. In particular, the class of saline reedbeds (A2.53C), the surface area of which increased significantly in the period of observation (PLAND), also showed a higher landscape complexity, as proved by the increasing values of NP, PSSD, ED, SHAPE and PD. There are contrasting trends for the metric values associated with class I1.3, which underwent a considerable decrease in the period of observation.

Figures 14 and 15 graphically display the values of the landscape metrics for CE, compared for the two years of observation. A gap appears for class C1.2, for which the results were lost in 2014. The most striking variation concerns the two classes E1.6 and I1.3; in fact, an almost total conversion of the first to the second can be observed, with an opposite trend in almost all metrics. As mentioned in Section 3.2.2, it is worth noting the COHESION values for classes A2.526 and A2.53D: variations in aggregation/distribution (clumpiness) correspond well to variations in surface area (CA, Table 4).

In Figure 16 the magnitude of changes of landscape metrics for the set of EUNIS classes shared between the two sites are compared. The main outcomes are:

- SHAPE showed opposite variations in A2.522 (Mediterranean Juncus sp.pl. salt-marshes), with a steep increase at the CE site and a reduction at the SC site; in both cases, class A2.522 suffered a reduction in CA, but while in SC the class was reduced

to a few regular patches, in CE it turned out fragmented into numerous patches, elongated in parallel to the coast line and very irregular in shape;

- class B1.1 (sand beach driftlines) has a natural elongated shape along the coastline; in this case, the SHAPE increment in CE was related to a reduction and fragmentation of this habitat type.;
- COHESION showed a steep decrease in class A2.526 (saltmarsh shrubs) in CE, related to a reduction in patch numbers, resulting in a marked spatial dispersion; the same applied for class J5.4.
- PD showed a steep decrease in class I1.3 in site SC: the reduction in surface area complemented the marked spatial dispersion in the landscape.

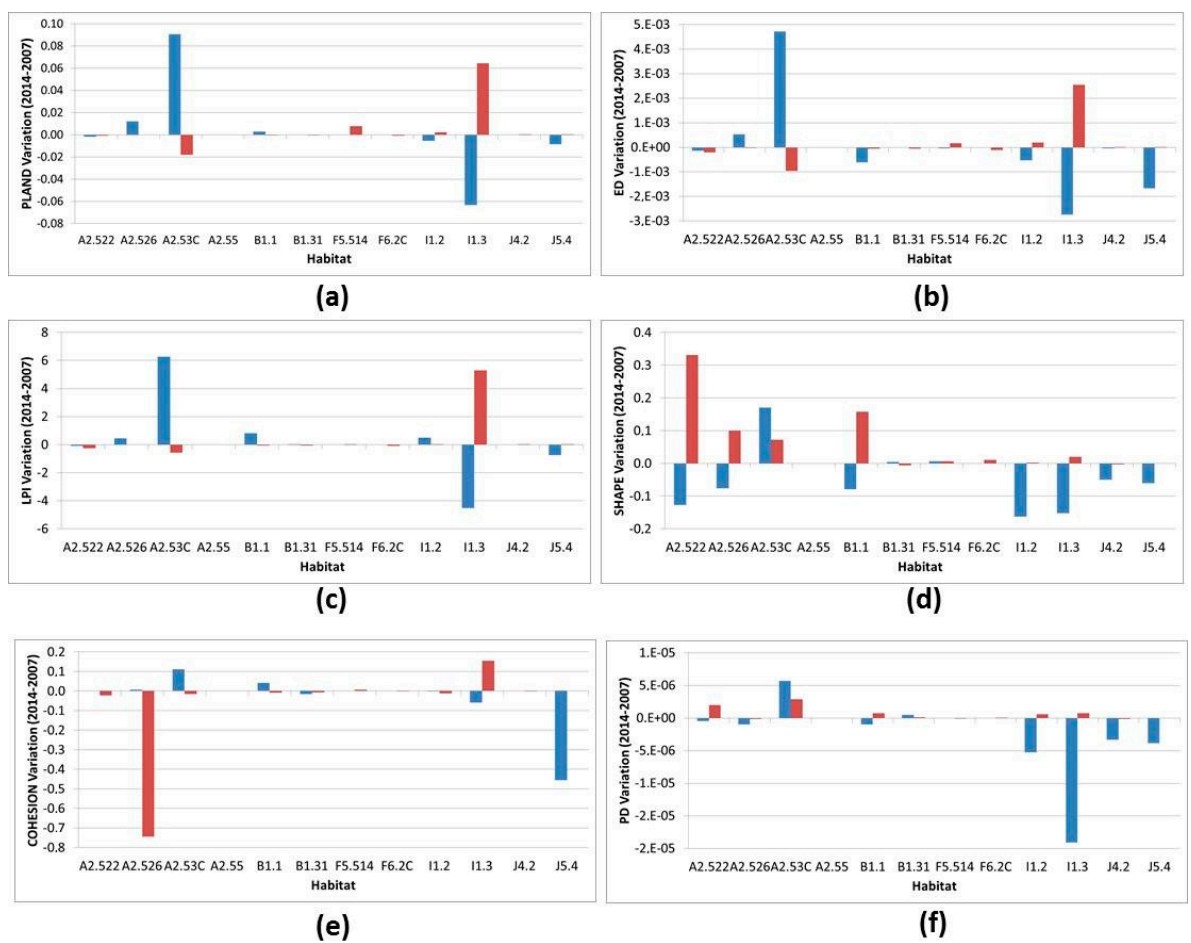

**Figure 16.** Barplots comparing the MCs of landscape metrics ((**a**) PLAND; (**b**) ED; (**c**) LPI; (**d**) SHAPE; (**e**) COHESION; (**f**) PD) for common EUNIS classes in the two sites SC (blue) and CE (red).

In Figure 17, landscape metrics at the landscape level for the two sites and in the two years of observation, calculated on the basis of the EUNIS map, are compared.

NP and PRD increased at the SC site, possibly due to the higher percentage of area change (as seen in Section 3.2.1). NP, PRD, MPS and PSSD showed opposite trends in the two sites, which would imply opposite trends in landscape complexity, as confirmed by SIDI (Figure 17g). In fact, the Shannon index is strongly influenced by class richness and by rare types, so it is very sensitive to even small diversity changes and has greater importance in interpreting the landscape; the Simpson index gives more weight to evenness and common or dominant types and it can be used to show the trend of ecosystem diversity [41,47]. The raise of both SIDI and SHDI in the SC site could be due to both the increase in habitat types (more classes) and, at the same time, the marked and rapid expansion of reedbeds (A2.53C) and annual grasslands (E1.6) that have become dominant types in the landscape,

with a decrease in landscape complexity. In the CE site, no major changes were highlighted, except for the decrease in NP and an increase in PSSD.

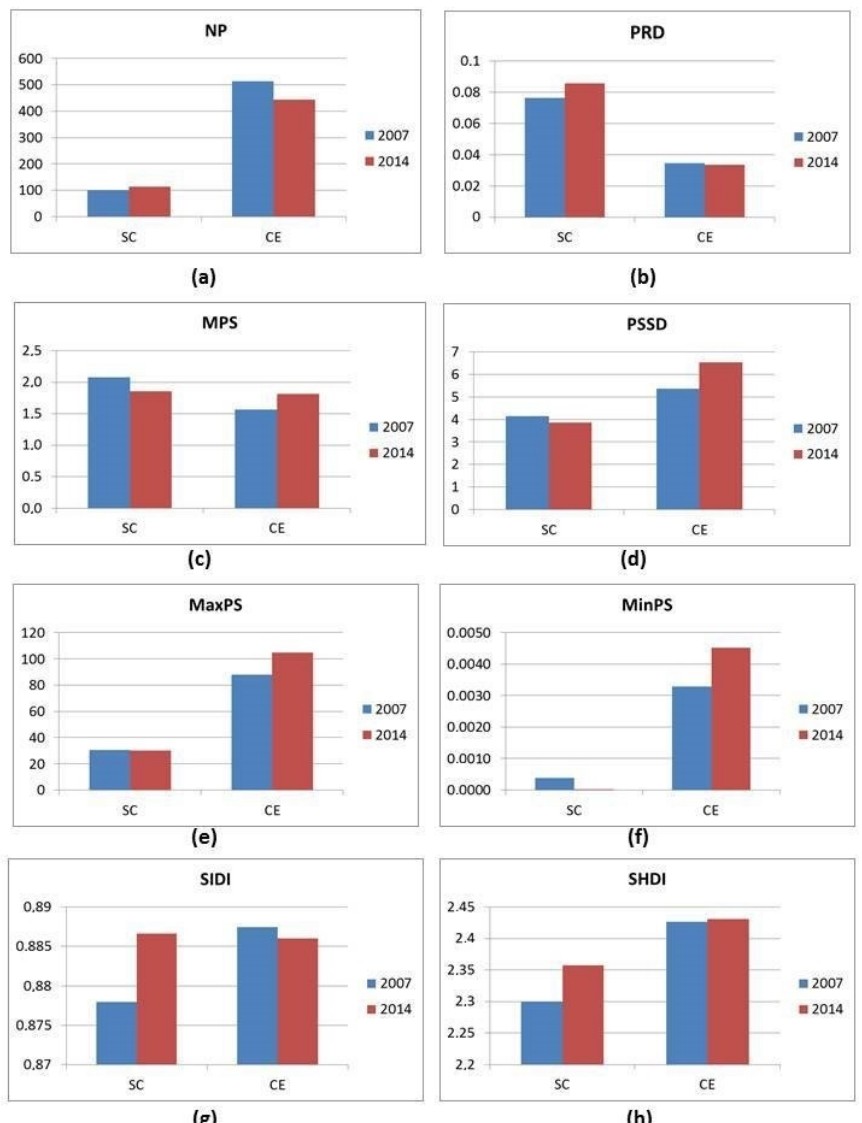

**Figure 17.** Barplots comparing the landscape metrics at landscape level ((**a**) NP; (**b**) PRD; (**c**) MPS; (**d**) PSSD; (**e**) MaxPS; (**f**) MinPS; (**g**) SIDI; (**h**) SHDI), calculated on the basis of the EUNIS map, at the two sites in the two years of observation.

*3.3. Impacts*

In Appendix B, a list of the impacts affecting natural and seminatural habitat types in the study sites is reported. For each specific type of impact (or stress) the following are included: the biodiversity target, a short description of the impact, broad categories of observed impacts, proximate pressures, underlying factors, inter-relations with other impacts and the habitat/LC type corresponding to the biodiversity target (according to EUNIS and LCCS taxonomies).

In both the study sites coastal erosion determines reduction or alteration drift lines, sometimes with conversion to bare sands or other unconsolidated materials; moreover, coastal erosion determines a wide range of stresses on biotic and abiotic systems and on numerous habitat types. In the case of the CE site, the progressive reduction of the sandbank has caused, over time, frequent breaks of the dune belt with an increasing inflow of marine waters and progressive salinization of the lagoons and of the associated environments. As a

consequence, habitat types characterized by Phragmites australis (marine reedbeds, EUNIS A2.53C) have been undergoing, over time, increasing death rate and class conversion to other types (e.g., halophytic communities or coastal lagoons) has been observed. In the SC site the opposite event is observed: Phragmites australis is rapidly spreading in the area, favored by fire and/or water eutrophication, and is causing alterations of other natural salt marshes communities (e.g., perennial glassworts).

Agriculture is another important driver of changes. The intensification of agricultural practices (especially uncontrolled practices) both inside and outside the protected areas has caused alterations. In SC there is widespread crop abandonment, with many agricultural areas, especially arable lands, converted into abandoned areas or reedbeds; but an intensification of agricultural areas can be observed close to the coastline in sensitive areas of the coastal belt, with a dramatic reduction of the annual glasswort communities (habitat A2.55). The agricultural areas are also a source of pollutants that determine the modification, over time, of the water quality of the coastal lagoons.

As regards sclerophyllous shrub vegetation—that is, Mediterranean maquis and garrigues (habitat types F5.514 and F2.C)—it is worth noting that no significant changes were recorded in the period of observation in terms of class area, which remained more or less the same, except for slightly significant variations. Therefore, these two classes did not show relevant class conversion but underwent major structural changes in both the two areas, as well as changes in terms of species composition. In particular, the encroachment of woody or other native shrubby plants affected the sclerophyllous shrub communities in both the study sites. In the CE site a spreading and thickening of Pinus halepensis into Mediterranean maquis and garrigues has been recorded and this rapidly increased in recent years, probably favored by frequent fires and by general climate change. Frequent fires also determined a change in the canopy cover of pine woods, which decreased over time. High fire frequency and overgrazing have caused, in the SC site, an encroachment of sprouting species such as Cistus sp. pl. and Calicotome infesta in garrigues dominated by the endemic Erica forskalii, as well as a thinning of the density of the Mediterranean maquis. All these changes, observed in the two sites to the detriment of sclerophyllous shrub vegetation, can be classified as class modifications (Appendix B).

## 4. Discussion

### 4.1. Change Representation

As pointed out in the previous sections, many changes have been identified in terms of both conversion and modification. As regards their cartographic representation, conversions (e.g., salt marsh vegetation to water; natural areas to cultivated) can be easily represented in the mapping process by simply shifting from one class to another. Modifications (e.g., change in the canopy cover of pine woods) cannot be represented.

Two instances of class modification detected in the surveyed areas are (a) the change in canopy cover (pine woods, Mediterranean maquis) and (b) the encroachment of Mediterranean maquis and garrigue. In such cases, with EUNIS, conversion is not involved and modification cannot be recorded and thus the same class was used in both 2007 and 2014. However, when using LCCS, the representation of some structural transformations is possible by changing the modifiers (Table 5). For the case of a change in canopy cover, in LCCS this modification can be easily represented by changing a classifier (the "cover" classifier) from A10 (closed) to A11 (open). The case observed in SC site, involved the decreasing (thinning) of the density and therefore of the cover of the Mediterranean maquis. In LCCS the original type is A12/A1.A4.A10.B3.XX.D1.E1-B9 (Broadleaved evergreen medium/high thicket) and it is possible, by simply changing the "cover" attribute (A12/A1.A4.A11.B3.XX.D1.E1-B9, Broadleaved evergreen medium/high shrubland), to describe the changed condition of this habitat type. A change in cover was also observed in the pine woods in the CE site, where the canopy cover decreased over time, from A12/A3.A10.B2.XX.D2.E1-B6 (Needleleaved evergreen medium high closed trees) to A12/A3.A11.B2.XX.D2.E1-B6 (Needleleaved evergreen woodland) (Table 5, Figure 18).

**Table 5.** A list of the different types of change at different levels of intensity affecting various types of forest vegetation observed in the two study areas ("Specific type of impact" column). For each type of change/degree of intensity, the corresponding class type is reported for EUNIS and LCCS. As regards the woody vegetation types surveyed in this contribution, LCCS allows the representation of some structural changes (e.g., cover or encroachment).

| Specific Type of Impact | Level of Intensity | EUNIS | LCCS Code | LCCS Description |
|---|---|---|---|---|
| Change in cover (thinning) in Mediterranean maquis | High density | F5.514–Lentisc brush | A12/A1.A4.A10.B3.XX.D1.E1-B9 | Broadleaved evergreen medium/high thicket |
| | Low density | | A12/A1.A4.A11.B3.XX.D1.E1-B9 | Broadleaved evergreen medium/high shrubland |
| Change in canopy cover in pine forests | High density | G3.F1–native conifer plantations | A12/A3.A10.B2.XX.D2.E1-B6 | Needleleaved evergreen medium/high closed trees |
| | Low density | | A12/A3.A11.B2.XX.D2.E1-B6 | Needleleaved evergreen woodland |
| Pine encroachment in Mediterranean garrigues | Original garrigue | F6.2C–Eastern Erica garrigues | A12/A1.A4.A11.B3.D1.E1/B10 | Broadleaved evergreen open (65–15%) dwarf shrublands |
| | Low encroachment (sparse trees) | | A12/A1.A4.A11.B3.D1.E1.F2.F5.F10.G2/B10.G7 | Broadleaved evergreen open dwarf shrublands with low emergents |
| | Pine encroachment of 15–65% | | A12/A1.A4.A11.B3.D2.E1.F2.F6.F7.G3/B9.F9.G10 | Needleleaved evergreen open (65–15%) medium/high shrubland with open dwarf shrubs |
| | Pine encroachment of >65% | G3.F–native conifer plantations | A12/A1.A4.A10.B3.D2.E1. F2.F6.F7.G3/ B9.F9.G10 | Needleleaved evergreen closed (>65%) low trees with open dwarf shrubs |
| Pine encroachment in Mediterranean maquis | Original maquis | F5.514–Lentisc brush | A12/A1.A4.A10.B3.XX.D1.E1-B9 | Broadleaved evergreen medium/high thicket |
| | Pine encroachment of 15–65% | | A12/A1.A3.A11.B2.XX.D2.E1.F2.F6.F7.G3-B7F8G9 | Needleleaved evergreen open low trees (woodland) with closed medium/high shrubs |
| Shrub encroachment in Mediterranean maquis | Original garrigue | F6.2C–Eastern Erica garrigues | A12/A1.A4.A11.B3.XX.D1.E1-B10 | Broadleaved evergreen open dwarf shrubland |
| | Low encroachment (sparse shrubs) | | A12/A1.A4.A11.B3.XX.D1.E1.F2.F6.F10.G3-B10.G9 | Broadleaved evergreen dwarf shrubland with medium high shrub emergents |

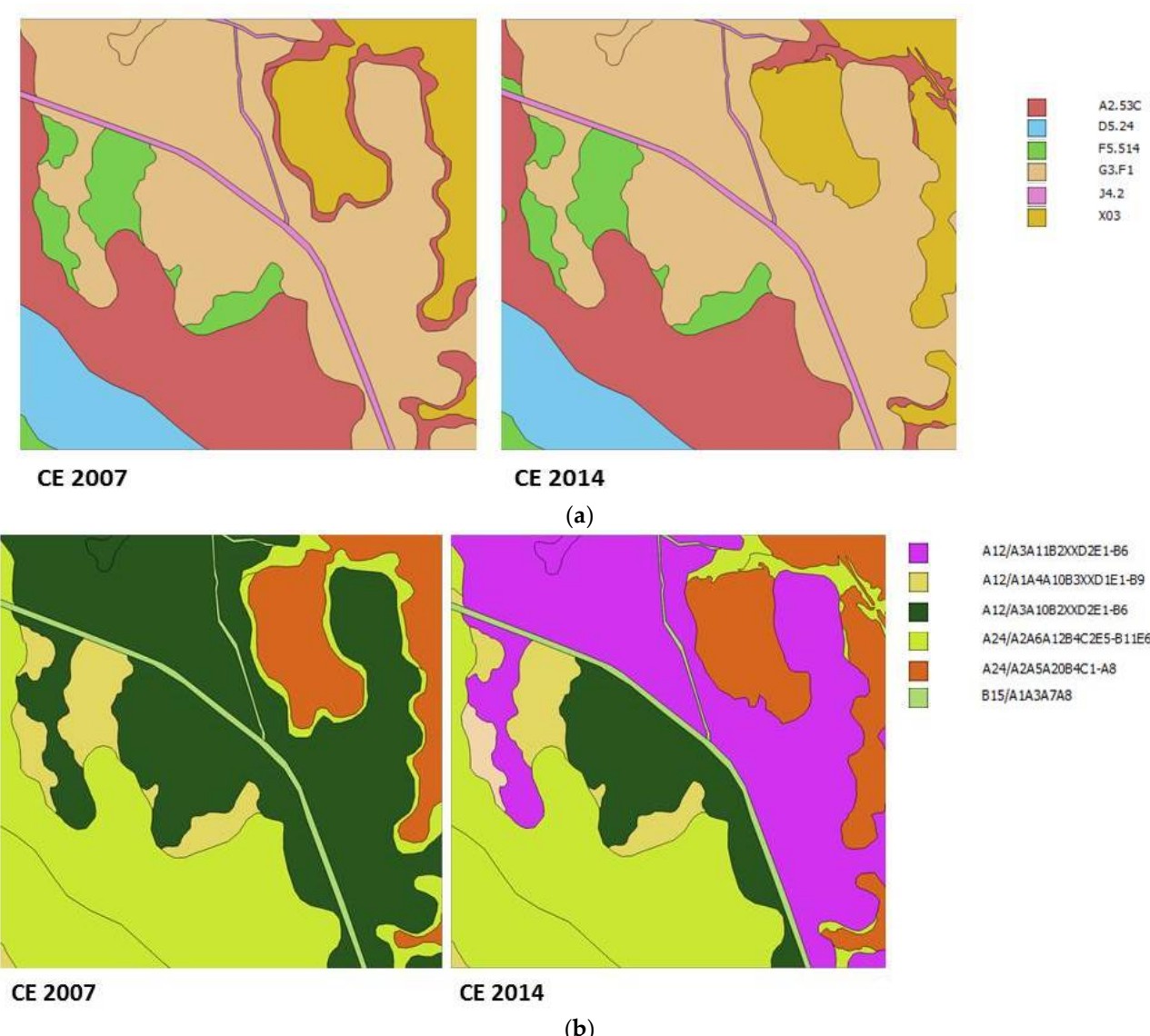

| | |
|---|---|
| ■ | A2.53C |
| ■ | D5.24 |
| ■ | F5.514 |
| ■ | G3.F1 |
| ■ | J4.2 |
| ■ | X03 |

CE 2007 · CE 2014

(**a**)

| | |
|---|---|
| ■ | A12/A3A11B2XXD2E1-B6 |
| ■ | A12/A1A4A10B3XXD1E1-B9 |
| ■ | A12/A3A10B2XXD2E1-B6 |
| ■ | A24/A2A6A12B4C2E5-B11E6 |
| ■ | A24/A2A5A20B4C1-A8 |
| ■ | B15/A1A3A7A8 |

CE 2007 · CE 2014

(**b**)

**Figure 18.** Changes in cover were observed in the pine woods in the CE site, where the canopy cover decreased over time. (**a**) The same EUNIS type, G3.F1 (native conifer plantations) was used in 2007 and in 2014; (**b**) the cover change of some areas is well represented with LCCS, changing from A12/A3.A10.B2.XX.D2.E1-B6 (Needleleaved evergreen medium high closed trees) to A12/A3.A11.B2.XX.D2.E1-B6 (Needleleaved evergreen woodland).

The representation of encroachment turned out to be more challenging, either because it occurred at various degrees of intensity or because of limitations inherent to the structure of LCCS. In the case of the pine encroachment in sclerophyllous shrub vegetation observed in the CE site, this could be differently treated and represented depending on the intensity. With EUNIS, if the impact is of slight intensity, the original typology (e.g., garrigues) is maintained (no change of representation); if it is of high intensity, with dense encroachment, it can be represented as a class conversion, e.g., from garrigues to pine woods. In LCCS it is possible to represent this type of change as a modification of the original class by adding a second layer. In the case of slight intensity (cover of the pine canopy less than 15%), the class A12-A1.A4.A11.B3.D1.E1/B10 (Broadleaved evergreen open (65–15%) dwarf shrublands) can become A12-A1.A4.A11.B3.D1.E1.F2.F5.F10.G2/B10.G7 (Broadleaved evergreen open dwarf shrublands with low emergents). In cases of higher intensity, a second layer of pine trees with higher cover should be specified. However, this is not allowed in the current

version of LCCS. LCCS allows the specification of a second layer "trees" on a main layer "shrubs" only if the trees have a cover less than 15% (sparse or emergent). It is not possible to specify garrigues or maquis as the principal layer and the trees as the second layer if the trees have a cover more than 15% (that is, if they change from open to closed). Thus, to represent this change, we were obliged to apply a class conversion, describing the class as pine trees for the main layer and garrigue (or maquis) for the second layer. Thus, the class A12-A1.A4.A11.B3.D1.E1/B10 (Broadleaved evergreen open (65–15%) dwarf shrublands) can turn into A12-A1.A4.A11.B3.D2.E1.F2.F6.F7.G3/B9.F9.G10 (Needleleaved evergreen open (65–15%) medium/high shrubland with open dwarf shrubs) or, in cases of higher intensity, into A12-A1.A4.A10.B3.D2.E1.F2.F6.F7.G3/B9.F9.G10 (Needleleaved evergreen closed (>65%) low trees with open dwarf shrubs). In both cases, pine wood becomes the first layer and the garrigue the second layer and therefore the user is obliged to represent the change as a class conversion, even if details about the stratification can be included. The case of pine encroachment in Mediterranean maquis is similar, with the original class A12/A1.A4.A10.B3.XX.D1.E1-B9 (Broadleaved evergreen medium/high thicket) changing into A12/A1.A3.A11.B2.XX.D2.E1.F2.F6.F7.G3-B7F8G9 (Needleleaved evergreen open low trees (woodland) with closed medium/high shrubs) (Figure 19; Table 5).

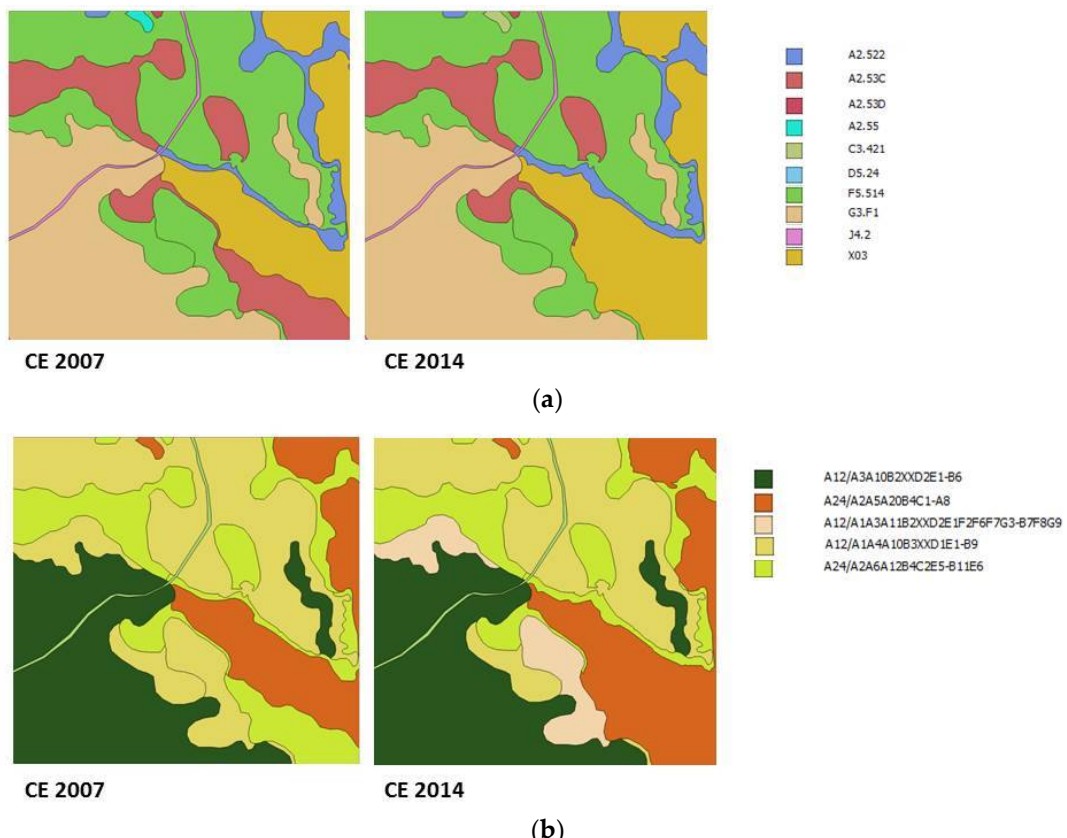

**Figure 19.** The case of pine encroachment in Mediterranean maquis in the CE site. In this specific case, being an encroachment of low intensity, the same EUNIS class (F5.514—Lentisc brush) was used in 2007 and in 2014 (**a**). With LCCS (**b**), the original class A12/A1.A4.A10.B3.XX.D1.E1-B9 (Broadleaved evergreen medium/high thicket) changed into A12/A1.A3.A11.B2.XX.D2.E1.F2.F6.F7.G3-B7F8G9 (Needleleaved evergreen open low trees (woodland) with closed medium/high shrubs).

In the SC site the Erica garrigue (A12/A1.A4.A11.B3.XX.D1.E1-B10 (Broadleaved evergreen open dwarf shrubland)) was prone to encroachment by some sprouting shrub species, such as Cistus sp.pl. and Calicotome infesta. In this case, because of the sparse cover of the invasive species, it is possible to specify in LCCS a second layer of emergents, maintaining the garrigue as the main layer (A12/A1.A4.A11.B3.XX.D1.E1.F2.F6.F10.G3-

B10.G9 (Broadleaved evergreen dwarf shrubland with medium/high shrub emergent)). These and other cases are summarized in Table 5.

As pointed out in Section 3.2.2, LCCS does not allow the discrimination of the various classes of salt marsh vegetation (notably with regard to different types of helophyte vegetation, i.e., rushes, sedges and reeds), due to the scarce thematic resolution of the class A24/A1A4A12B3C2D3-B10, nor the detection of changes within this class (that is, intra-class changes), as already highlighted in previous studies [9,21,23]. In such cases, EUNIS, with its detailed list of vegetation types, is fundamental for discrimination between these classes, as illustrated in Figure 20.

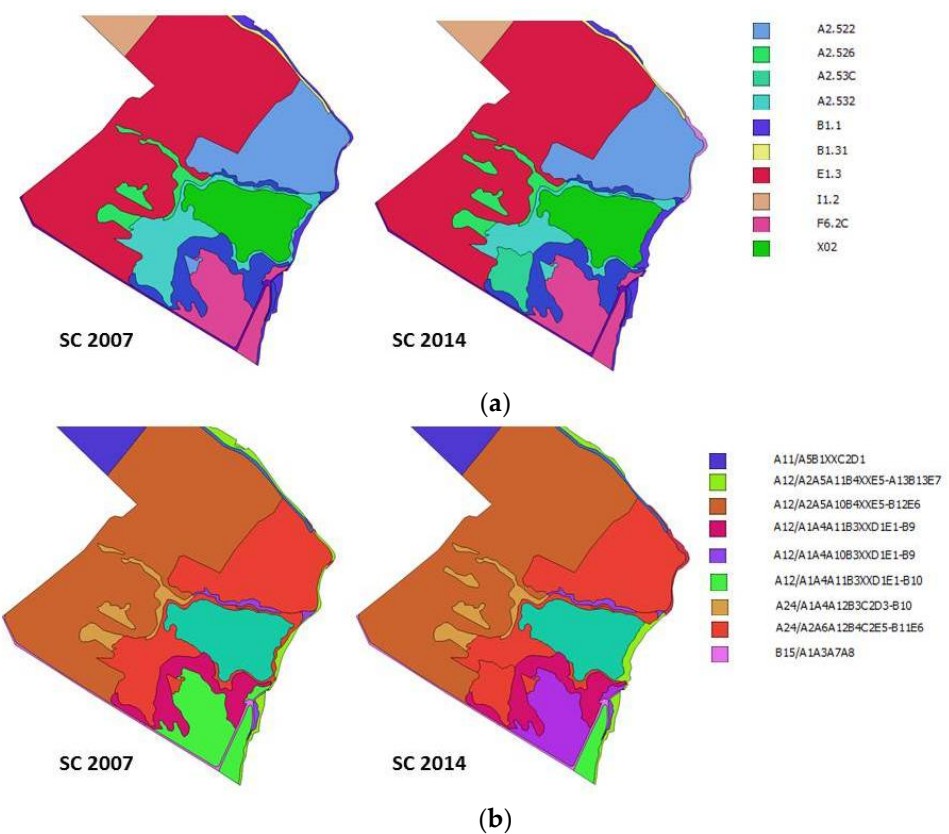

**Figure 20.** In LCCS (**b**), helophytes are treated as a unique class (A24/A2A6A12B4C2E5-B11E6 (Perennial closed tall grasslands on temporarily flooded land)). In such cases, EUNIS (**a**), with its detailed list of vegetation types, becomes fundamental for discrimination among these classes.

Thus, if LCCS allowed the detection of the various class modifications that affected large areas of woody vegetation (pine woods, maquis and garrigues) in both sites, EUNIS made it possible to better focus on the different types of salt marsh vegetation (especially helophyte vegetation, i.e., rushes, sedges and reeds), some of which were subject to intense reduction in LS.

### 4.2. Landscape Metrics

One general outcome detected in this study was a change in landscape composition and configuration, at the class level, expressed by the variations in landscape metrics in both the sites, especially in those classes that were subject to major changes in class area (e.g., salt marshes A2.53C, A2.532 and A2.55 or agricultural-managed areas E1.6 and I1.3). As was expected, there was no univocal trend for similar or related classes within the same area and not even for the same classes in the two sites. In most cases, a reduction in habitat did not imply an increase in its spatial complexity; on the contrary, as the surface decreased, landscape complexity decreased as well. This particular behavior is related to both the

small size of the study areas and their high landscape complexity. COHESION values were quite high for almost all classes, especially in natural habitats with a degree of "clumpiness" (that is, fragmented but with the tendency to form aggregates in more or less limited areas, such as, for example, F5.514 and F6.2C in the CE site or A2.526 and A2.532 in the SC site).

It is worth noting that different thematic resolutions related to the use of different taxonomies affects spatial pattern and landscape metrics performances [48–50]. Varying the thematic resolution by shifting the landscape classification scheme affects the patchiness of mosaics representing natural landscapes and has considerable effects on class-level metrics [9].

### *4.3. Site Management*

One of the main causes of failure of management activities and policy intervention in wetlands is a lack of consistency among government policies (economics, environment, nature protection, physical planning, etc.) applied in different sites, even when they belong to the same administrative area or region [51,52], whereas a sustainable use of wetlands requires management approaches based on interconnections between different areas.

Wetlands in Southern Italy represent an endangered and invaluable ecological heritage. Most of these wetlands are protected by various management plans and conservation programs. Even when they belong to the same biogeographic areas, these wetlands often have very different conservation statuses as well as being subject to different policies presently and in the past. Among the main pressures detected, fire (climate change), coastal erosion (uncontrolled building in surrounding areas), water salinization (changes in hydrodynamics and intensification of agricultural practices), water eutrophication (agricultural practices intensification) are among the most relevant. In particular, the Common Agricultural Policy (CAP) and local policies are mostly responsible for changes in land management over the past 40 years. It is now widely recognized that these agricultural policies do not sufficiently consider the relationships between agriculture and environment, in spite of the conservation constraints introduced with the creation of protected areas and the Natura 2000 network. The majority of coastal wetland areas in Southern Italy are "nested" in a matrix of agricultural landscapes; agricultural areas are widespread outside and within the boundaries of protected areas. Arable lands have frequently been extended up to the edges of marshes and numerous natural habitats have been affected by this process. Thus, the adoption of proper policies in agriculture, inside and outside protected areas, and the deployment of surveillance actions are key elements for correct management. In the cases of the two study sites, contrasting trends were observed: in SC arable lands were, in large part, converted into fallow lands, partly due to land abandonment and partly due to crop rotations; in CE the opposite trend was observed, with part of the fallow lands and seminatural grasslands converted into arable lands.

As regards the habitat modifications affecting some forest habitats (e.g., encroachment or change in cover in classes F5.514, F6.2C and, G3.F1), these were mostly due to fire; this pressures seemed to indifferently affect the two areas, more or less with the same severity. This process, as well as coastal erosion, does not seem to depend on local management policies but on policies taken at a wider regional or global level.

### 5. Conclusions

The results of our observations confirm that the surveyed coastal sites are subject to changes and that these changes are occurring very quickly. Nevertheless, considering the small area size and the limited time period of the observations, the ecosystems concerned and related services have not yet been compromised; further mapping activities will make it possible to verify whether the detected trends will be mitigated or not and whether ecosystem function and services will be altered. The ongoing dynamics, mainly related to anthropogenic activities (agricultural above all), are made even more complex by the already existing complexity of the territory, heritage of a very long and ancient history of land use. As with almost everywhere in the Mediterranean area, in order to understand the

current dynamics it is necessary to know the land-use history of the site. Different land-use traditions lead to the development of different dynamics and to different levels of conservation. The same pressures can also lead to different effects under different management policies. In detecting and monitoring these processes, it is crucial to use, in addition to a correct spatial scale of observation, the appropriate thematic resolution and an appropriate taxonomy (or more than one). While EUNIS allows the detailed definition of many natural and seminatural habitat types, LCCS allows the identification and representation of intraclass modifications. For long term habitat monitoring and change detection, the coupling of habitat taxonomies such as EUNIS and LCCS is recommended. LCCS has some limits, e.g., in the description of main and second layers in some cases of encroachment; however, these issues may be addressed in the future in view of enhancing LCCS as a tool for long term monitoring.

**Author Contributions:** Conceptualization, V.T. and M.A.; methodology, V.T. and M.A.; validation, V.T. and G.V.; formal analysis, G.V. and M.A.; writing—original draft preparation, V.T., G.V. and M.A.; writing—review and editing, V.T. and M.A.; funding acquisition, V.T. and M.A. All authors have read and agreed to the published version of the manuscript.

**Funding:** This research was funded by the INTERREG IIIA Greece-Italy INFO-NAT project, by the H2020 E-SHAPE project—EuroGEO Showcases: Applications Powered by Europe (www.e-shape.eu), Grant Agreement: 820852 and by the COHECO project—Sistema Integrato di757 Monitoraggio, Allerta e Prevenzione dello stato di COnservazione di Habitat ed ECOsistemi in aree interne e costiere protette e da proteggere (www.coheco.it) in the framework of the POR Puglia FESR-FSE 2014–2020.

**Acknowledgments:** The authors thank the three anonymous reviewers and the editor for their very useful comments and suggestions on the previously submitted manuscript.

**Conflicts of Interest:** The funders had no role in the design of the study; in the collection, analyses, or interpretation of data; in the writing of the manuscript, or in the decision to publish the results.

# Appendix A

**Table A1.** Reference table showing the complete list of habitat types in EUNIS and of LC classes in LCCS with the corresponding descriptions.

| EUNIS | EUNIS DESCRIPTION | LCCS | LCCS DESCRIPTION |
|---|---|---|---|
| J2.1 | Scattered residential buildings | B15/A1A4A13A17 | Scattered urban areas |
| J2.2 | Rural public buildings | B15/A1A4A12A17 | Scattered industrial and/or other areas |
| J2.4 | Agricultural constructions | | |
| J2.7 | Rural construction and demolition sites | | |
| J1.4 | Urban and suburban industrial and commercial sites | | |
| J4.2 | Road networks | B15/A1A3A7A8 | Paved roads |
| G5.5 | Small mixed broadleaved and coniferous anthropogenic woodlands | A11/A1C2D1-A7A9 | Rainfed broadleaved evergreen tree crops |
| I1.3 | Arable land with unmixed crops grown by low-intensity agricultural methods | A11/A3A4B1XXC1D1 | Monoculture of large- to medium-sized fields of rainfed graminoid crops (single crop) |
| | | A11/A3A4B2XXC1D1 | Monoculture of small-sized fields of rainfed graminoid crops (single crop) |
| I1.2 | Mixed crops of market gardens and horticulture | A11/A5B2XXC2D3 | Small-sized fields of irrigated nongraminoid crops |
| G2.91 | Olea europaea groves | A11/A1B1XXC1D1W8-A7A9B4 | Monoculture of medium-sized fields of broadleaved evergreen tree crops (orchards) |
| G3.F1 | Native conifer plantations | A12/A3A10B2XXD2E1-B6 | Needleleaved evergreen medium/high trees |
| | | A12/A3A11B2XXD2E1-B6 | Needleleaved evergreen woodland |
| E1.3 | Mediterranean xeric grassland | A12/A2A5A10B4XXE5-B12E6 | Closed perennial medium/tall forbs |
| F5.51 | Thermo-Mediterranean brushes, thickets and heath-garrigues | A12/A1A4A10B3XXD1E2-B9 | Broadleaved deciduous closed medium/high shrubland |

Table A1. *Cont.*

| EUNIS | EUNIS DESCRIPTION | LCCS | LCCS DESCRIPTION |
|---|---|---|---|
| F6.2C | | A12/A1A4A11B3XXD1E1-B10 | Broadleaved evergreen open dwarf shrubland |
| | Eastern Erica garrigues | A12/A1A4A11B3XXD2E1F2F6F7G3-B9F9G10 | Needleleaved evergreen open (65–15%) medium/high shrubland with open dwarf shrubs |
| | | A12/A1A4A11B3XXD1E1F2F6F10G3-B10G9 | Broadleaved evergreen dwarf shrubland with medium/high shrub emergents |
| F5.514 | Lentisc brush | A12/A1A4A10B3XXD1E1-B9 | Broadleaved evergreen medium/high thicket |
| | | A12/A1A4A11B3XXD1E1-B9 | Broadleaved evergreen medium/high shrubland |
| | | A12/A1A3A11B2XXD2E1F2F6F7G3-B7F8G9 | Needleleaved evergreen open low trees with closed medium/high shrubs |
| B1.1 | Sand beach driftlines | A12/A2A5A11B4XXE5-A13B13E7 | Open (40-(20-10)%) annual short forbs |
| B1.31 | Embryonic shifting dunes | A12/A2A6A11B4XXE5-A12B12E6 | Open ((70-60)-40%) perennial medium/tall grasslands |
| B1.631 | Dune prickly juniper thickets | A12/A1A4A10B3XXD2E1-B9 | Needleaved evergreen medium/high closed shrubland |
| B2.2 | Unvegetated mobile shingle beaches above the driftline | B16/A5-A12 | Stony bare soil and/or other unconsolidated materials |
| E1.6 | Subnitrophilous annual grassland | A12/A2A5A10B4XXE5-B12E7 | Closed annual medium/tall forbs |
| E1.313 | Mediterranean annual communities of shallow soils | A12/A2A5A11B4XXE5-A13B13E7 | Open (40-(20-10)%) annual short forbs |
| A2.526 | Mediterranean saltmarsh scrubs | A24/A1A4A12B3C2D3-B10 | Aphyllous closed dwarf shrubs on temporarily flooded land |
| A2.55 | Pioneer saltmarshes | A24/A2A5A13B4C2E5-B13E7 | Open annual short herbaceous vegetation on temporarily flooded land |
| A2.522 | Mediterranean Juncus maritimus and Juncus acutus saltmarshes | A24/A2A6A12B4C2E5-B11E6 | Perennial closed tall grasslands on temporarily flooded land |
| A2.532 | Mediterranean halo-psammophile meadows | | |
| A2.53C | Marine saline beds of Phragmites australis | | |
| A2.53D | Geolitt wetlands and meadows: saline and brackish reed, rush and sedge stands | | |
| D5.24 | Fen (Cladium mariscus) beds | | |
| C1.2 | Permanent mesotrophic lakes, ponds and pools | A24/A5A20B4C2-A8B13 | Closed to open short rooted forbs on temporarily flooded land |

<div align="center">Table A1. <i>Cont.</i></div>

| EUNIS | EUNIS DESCRIPTION | LCCS | LCCS DESCRIPTION |
|---|---|---|---|
| C3.421 | Short Mediterranean amphibious communities | A24/A2A5A13B4C2E5-B13E7 | Open annual short herbaceous vegetation on temporarily flooded land |
| D5.1 | Reedbeds normally without free-standing water | A24/A2A6A12B4C2E5-B11E6 | Perennial closed tall grasslands on temporarily flooded land |
| J5.4 | Nonsaline water channels with completely man-made substrate | B27/A1B1C1A4 | Deep to medium artificial perennial waterbodies (flowing) |
| X02 | Saline coastal lagoons | A24/A2A5A13B4C1E5-A15B12E6 | Perennial open (40-(20-10)%) medium/tall forbs on permanently flooded land |
| X03 | Brackish coastal lagoons | | |

## Appendix B

**Table A2.** List of the impacts affecting natural and seminatural habitat types in the study sites. For each specific type of impact the following are included: biodiversity target, a short description of the impact, broad categories of observed impacts, proximate pressures, underlying factors, inter-relations with other impacts and the habitat/LC type corresponding to the biodiversity target (according to EUNIS and LCCS taxonomies). In some cases, impacts on abiotic systems can become direct threats to biotic systems (e.g., changes in the water regime of coastal lagoons that lead to the loss of annual glasswort communities (EUNIS A2.55) or changes in water quality (water salinization) in coastal lagoons that become direct threats to reed beds (loss of reed beds communities)).

| Biodiversity Target | Site | Specific Type of Impact (Stress) | Broad Impact Category (Nagendra et al. 2012) | Short Description of the Impact (stress) | Direct Threat (Proximate Pressure) | Underlying Factors | Inter-Relations with Other Impacts | Habitat and LC Types |
|---|---|---|---|---|---|---|---|---|
| Mediterranean maquis and relevant mosaics | CE | Pine encroachment in Mediterranean maquis | Land cover/habitat modification | In recent years a spread of Pinus spp. into the scrubs environments (maquis and garrigues) has been observed. Young pine plants development seems to be favored by fire and climate change. | Fire | Changes in fire regime (increase in fire frequency), climate change (temperature extremes) | Pine encroachment in Mediterranean garrigues/change in species composition | EUNIS F5.514 LCCS A12/A1A4A10B3XXD1E1-B9 and A12/A1A3A11B2XXD2E1F2F6F7G3-B7F8G9 |
| Mediterranean maquis and relevant mosaics | SC | Change in cover (thinning) in Mediterranean maquis | Land cover/habitat modification | In recent years a thinning of the density and therefore of the cover of the Mediterranean maquis has been observed. Most likely due to fire. | Fire | Changes in fire regime (increase in fire frequency) | Change in species composition | EUNIS F5.514 LCCS A12/A1A4A10B3XXD1E1-B9 and A12/A1A4A11B3XXD1E1-B9 |

Table A2. *Cont.*

| Biodiversity Target | Site | Specific Type of Impact (Stress) | Broad Impact Category (Nagendra et al. 2012) | Short Description of the Impact (stress) | Direct Threat (Proximate Pressure) | Underlying Factors | Inter-Relations with Other Impacts | Habitat and LC Types |
|---|---|---|---|---|---|---|---|---|
| Mediterranean garrigues and relevant mosaics | CE | Pine encroachment in Mediterranean garrigues | Land cover/habitat modification | In recent years a spread of *Pinus* spp. Into the scrubs environments (maquis and garrigues) has been observed. The growth of young pine plants seems to be favored by fire and climate change. | Fire | Changes in fire regime (increase in fire frequency), climate change (temperature extremes) | Pine encroachment in Mediterranean maquis/change in species composition | EUNIS F6.2C LCCS A12/A1A4A11B3XXD1E1-B10 and A12/A1A4A11B3XXD2E1F2F6F7G3-B9F9G10 |
| Mediterranean garrigues and relevant mosaics | SC | Shrub encroachment in Mediterranean garrigues | Land cover/habitat modification | Maybe due to both overgrazing and higher fire frequency, an encroachment of sprouting species such as *Cistus* sp. pl. and *Calicotome infesta* in garrigues dominated by *Erica forskalii* has been observed. | Ranching and grazing/fire | Lack of adequate monitoring and management measures in the area | Change in species composition | EUNIS F6.2C LCCS A12/A1A4A11B3XXD1E1-B10 and A12/A1A4A11B3XXD1E1F2F6F10G3-B10G9 |
| Pine plantations | CE | Change in canopy cover | Changes in plant community structure | Pine forests (old plantations) are threatened by arsons. The pine forest cover decreases over time. | Fire | Changes in fire regime (increase in fire frequency), climate change (temperature extremes) | Change in species composition | EUNIS G3.F1 LCCS A12/A3A10B2XXD2E1-B6 and A12/A3A11B2XXD2E1-B6 |
| Dune vegetation | CE SC | Habitat loss and fragmentation | Habitat fragmentation and changes in landscape connectivity | Coastal erosion and other anthropogenic pressures are determining loss and fragmentation of coastal dune habitat types, with changes in CA (class area), patch size, number and shape. | Structural changes in the hydrodynamic conditions—coastal erosion | Uncontrolled building on the lands surrounding the area | Change in species composition | EUNIS B1.1, B1.31, B1.32, B1.361 LCCS A12/A2A5A11B4XXE5-A13B13E7, A12/A2A6A11B4XXE5-A12B12E6, A12/A2A6A10B4XXE5-B11E6, A12/A1A4A10B3XXD2E1-B9 |
| Dune vegetation | SC | Change in species composition | Changes in plant community structure | Due to habitat fragmentation and also the intense tourist flow and trampling (especially in summer), several dune plant communities are undergoing changes in species composition. | Human intrusions and disturbance (recreational) | Lack of adequate monitoring and management measures in the area | Habitat loss and fragmentation | EUNIS B1.1, B1.31, B1.32 LCCS A12/A2A5A11B4XXE5-A13B13E7, A12/A2A6A11B4XXE5-A12B12E6, A12/A2A6A10B4XXE5-B11E6 |

**Table A2.** *Cont.*

| Biodiversity Target | Site | Specific Type of Impact (Stress) | Broad Impact Category (Nagendra et al. 2012) | Short Description of the Impact (stress) | Direct Threat (Proximate Pressure) | Underlying Factors | Inter-Relations with Other Impacts | Habitat and LC Types |
|---|---|---|---|---|---|---|---|---|
| Brackish marshes | CE | Loss of reed beds communities | Land cover/habitat conversion | Due to water salinization, an increasing death rate of reeds beds communities has been observed over time. | Change in water quality (water salinization) | Structural changes in the hydrodynamic conditions—coastal erosion | Change in water quality (water salinization) | EUNIS A2.53C LCCS A24/A2A6A12B4C2E5-B11E6 |
| Brackish marshes | CE | Conversion of *Cladium mariscus* communities (habitat 7210) to reed beds | Land cover/habitat conversion | Phragmites australis spreads easily after cutting or fire, causing alterations in other natural salt marshes communities (especially Cladium mariscus communities). | Problematic native species | Agro-forestry practices | Change in species composition | EUNIS D5.24, A2.53C LCCS A24/A2A6A12B4C2E5-B11E6 |
| Salt marshes | SC | Conversion of perennial glasswort communities to reed beds | Land cover/habitat conversion | Phragmites australis is rapidly spreading in the area, often causing alterations in other natural salt marshes communities. | Change in water regime | Agro-forestry practices | Change in species composition | EUNIS A2.526, A2.53C LCCS A24/A1A4A12B3C2D3-B10, A24/A2A6A12B4C2E5-B11E6 |
| Brackish marshes | CE | Loss of annual glasswort communities (habitat 1310) and conversion to coastal lagoons | Land cover/habitat conversion | The annual glasswort (*Salicornia patula*) communities are slightly reduced. | Change in water regime | Structural changes in the hydrodynamic conditions—coastal erosion | Change in water regime | EUNIS A2.55 LCCS A24/A2A5A13B4C2E5-B13E7 |
| Salt marshes | SC | Loss of annual glasswort communities (habitat 1310) and conversion to agricultural areas | Land cover/habitat conversion | The annual glasswort (*Suaeda* sp.pl., *Salsola* sp.pl., *Cressa cretica*) communities are dramatically reduced. | Farming | Lack of adequate monitoring and management measures in the area | | EUNIS A2.55 LCCS A24/A2A5A13B4C2E5-B13E7 |
| Coastal lagoons | CE | Change in water quality (water salinization) | Land cover/habitat modification | In LC, the higher inflow of marine water due to the frequent breaks of coastal dunes determines the salinization of waters of lagoons and related environments. | Structural changes in the hydrodynamic conditions—coastal erosion | Uncontrolled building on the lands surrounding the area | Change in water regime/loss of reed beds communities | EUNIS X03 LCCS A24/A2A5A13B4C1E5-A15B12E6 |

**Table A2.** *Cont.*

| Biodiversity Target | Site | Specific Type of Impact (Stress) | Broad Impact Category (Nagendra et al. 2012) | Short Description of the Impact (stress) | Direct Threat (Proximate Pressure) | Underlying Factors | Inter-Relations with Other Impacts | Habitat and LC Types |
|---|---|---|---|---|---|---|---|---|
| Coastal lagoons | CE | Change in water regime | Land cover/habitat modification | In CE, frequent breaks of coastal lead to a much higher inflow of marine water. | Structural changes in the hydrodynamic conditions—coastal erosion | Uncontrolled building on the lands surrounding the area | Change in water quality (water salinization)/loss of annual glasswort communities | EUNIS X03 LCCS A24/A2A5A13B4C1E5-A15B12E6 |
| Coastal lagoons | SC | Change in water quality (water pollution) | Land cover/habitat modification | The agricultural areas surrounding the area are a source of pollutants for the lagoon waters, determining the modification, over time, of vegetation and species composition. | Agricultural effluents | Agricultural practices intensification | | EUNIS X02 LCCS A24/A2A5A13B4C1E5-A15B12E6 |
| Temporary ponds | CE | Change in species composition | Land cover/habitat modification | In CE, temporary ponds are small isolated patches nested in a matrix of agricultural surface. Expansion of agricultural practices causes alterations in floristic composition. | Farming | Agricultural policies and incentives | | EUNIS C3.421 LCCS A24/A2A5A13B4C2E5-B13E7 |

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
