# Peer review of "Monitoring and Recording Changes in Natural Landscapes: A Case Study from Two Coastal Wetlands in SE Italy"

_land, doi:10.3390/land10010050_

Round 1

Reviewer 1 Report

Line 18: Space is needed for “whilethe”.

Line 19: Modification or modification?

Line 21 and later: Full names should be added for the first time since the authors from other countries may be not familiar with the European nature information system (EUNIS) and Land Cover Classification System (LCCS).

Line 92-Line 119: A classification name for each LC class can be represented in a way of a table or something like a note. As the strings are showed several places later, it can be helpful for readers to retrieve.

Line 143-Line 148: Some references are suggested to be added in the selection of the landscape matrix.

Figure 2 and Figure 3: One legend for each site. A scale is necessary.
Line 204-Line 210: The materials are related to the methods. I am wondering if these can be moved and merged with section 2.3.

Line 210: Formulas should be numbered.

Line 236: Typing error.

Line 237: Typing error.

For the change detection analysis. Some illustration maps (maybe 2-3 inches with medium to high-resolution satellite imagery) can be clear to authors about the detailed and interesting areas.

Figure 8: The unit is needed for the “Magnitude”.

Line 434: The background information for the Land Cover Classification System (LCCS) can be moved to the introduction.

Table 5: It can be re-arranged to place it on one page.

Also, I am wondering if the authors can update the map classification into recent years, to inform to the Stakeholders of the current situation.

Author Response

Dear Reviewer, thank you for your comments and suggestions. We have revised the manuscript accordingly.

Reviewer 2 Report

Authors have done tremendous job to carry this important research. It has high practical importance. However, before it can be accepted for publication, it must address following comments:

  1. As you have mentioned that one of the motivation to do this analysis as this area is rich in biodiversity and valuable from ecosystem services point of view. However, because of this small area size and short time period, do authors think that it will have a huge differences in ecosystem services? 
  2. This part needs further explanation. What kind of different threats and pressures?
  3. Explanation for different land cover land use classes for both study areas from figure 2 and 3 is very important to understand the whole story in this case. Please add detailed explanation here. Also, I can not find the what land cover these acronym symbolizes. Please clarify
  4. Please explain % cloud cover for different satellite images? Also there is no information about accuracy assessment for different image processing.
  5. Figure 4 to 8 has very low resolution and font size is too small to read clearly. Please replace them.

Author Response

(The authors gave the same response as above.)

Reviewer 3 Report

This study is impressive. It shows how the authors painstakingly mapped out the transition of the land understudied from 2007 to 2014 using appropriate mapping techniques. It shows how land vegetation cover changes quickly, the drivers of these changes, and how forest management policies come into play in mitigating such pressures.  These are the kind of decision support systems that we need to support policymakers and forest managers. While the study has its own merits to be published, I think it also has more room for improvement. Hereunder are my comments.  

LINE/S

COMMENT/S

General

How do you handle such huge discrepancies and spikes/outliers (as shown in Figure 9)? What are the sources of error or variations and the level of confidence in your study? Maybe caution the readers about the accuracy of your results.

The study is reckoned from 2007 -2014 land cover. Is there a plan to continue this into a more recent data say, 2019 or 2020 at least? There must be a lot of changes occurring in the span of 6 years. Please justify why the study was not done utilizing a more recent land cover.    

15-16

Spell out add “(LC)” after Land Cover as you abbreviated it in line 16.

19

‘whilethe’ put a space between after ‘while’

20

What is ‘LCCS’

21

What is EUNIS… sentence is vague

64

‘doesn’t’ into ‘does not’

174

Change ‘2’ into ‘two’

Map legends

I think the map would be more understandable if the legend name would be renamed into something more sensible. Example, codenames of land covers such as A2.55 would be renamed into saltmarsh…and so on (e.g.  grassland, saline, pine forest, etc.)  Do this also in all tables and throughout the manuscript. Codenames are meaningless if one cannot imagine what that code is.

236

Remove ‘(‘ before ‘.’

Figure 8

Is this a histogram or just a barplot?

Figure 12

X and y axes labels were too small, as well as the legends

598

‘as regards’ into ‘As regards’

650

Remove another ‘.’ At the end of the sentence

Author Response

(The authors gave the same response as above.)
